# Relationships between growth mindsets and math achievement across socioeconomic status in 74 countries: Evidence from PISA 2022

Pimmada Charoensilp[1], Hanjoe Kim [2]*, Suppanut Sriutaisuk[1]*

**1** Faculty of Psychology, Chulalongkorn University, Bangkok, Thailand, **2** Department of Psychology, Yonsei University, Seoul, South Korea

* suppanut.s@chula.ac.th (SS); hanjoekim@yonsei.ac.kr (HK)

## Abstract

Growth mindsets have gained attention from education stakeholders due to their potential to enhance academic outcomes. However, some studies found that the relationship between mindsets and student achievement varied by socioeconomic status, while others found no association. As a result, it remains unclear whether growth mindsets are more beneficial for students from higher- or lower-socioeconomic backgrounds, or whether their benefits are consistent across all socioeconomic groups. This paper aims to investigate the moderating role of socioeconomic status in the relationship between growth mindsets and math achievement among 15-year-old students, using data from PISA 2022, which includes 507,588 representative students from 74 countries. The results showed that growth mindsets were significantly and positively related to achievement beyond socioeconomic status in the majority of countries. However, the moderating role of socioeconomic status varied across countries, with no consistent pattern emerging. These findings underscore the importance of considering contextual and cultural factors when promoting growth mindsets, as their benefits for math achievement may differ depending on students' socioeconomic backgrounds and the educational environments in which they are situated.

## Introduction

Over the past few decades, the concept of growth mindsets has emerged as a widely discussed concept in the fields of education, motivation, and psychology [1,2]. A growth mindset refers to the belief that certain attributes, such as intelligence, can be developed through effort, effective strategies, and help from others. In contrast, a fixed mindset refers to the belief that these attributes are innate and cannot be developed [3]. Research has linked growth mindsets to increased academic motivation, achievement, and well-being, particularly in the face of setbacks [4,5]. Although some controversy surrounds mindset research, recent findings suggest that the effects of

**Data availability statement:** All cleaned data files are available from the Open Science Framework database (https://osf.io/b2a6k/?view_only=9a352016b926445f-8130b5ad18eb9d56).

**Funding:** This research project is supported by Chulalongkorn University, the Second Century Fund (C2F), and has been granted funds from East West Psychological Science Research Center, Faculty of Psychology, Chulalongkorn University. The funders had no role in study design, data collection and analysis, decision to publish, or preparation of the manuscript.

**Competing interests:** The authors have declared that no competing interests exist.

growth mindsets on academic achievement may vary across contexts and populations [6]. In other words, growth mindsets tend to be particularly beneficial for individuals who are theoretically expected to gain from them [7]. Socioeconomic status is one such factor hypothesized to influence the extent to which students benefit from growth mindsets [8]. However, there is a limited number of large-scale, cross-national studies that have investigated this interaction.

Therefore, the main purpose of this paper is to examine the moderating role of socioeconomic status in the relationship between growth mindsets and math achievement among 15-year-old adolescents across 74 countries (or economies) using data from the Programme for International Student Assessment (PISA) 2022, a large-scale international assessment conducted by the Organisation for Economic Co-operation and Development (OECD).

**Controversial evidence in mindset research**

In the school context, mindsets are proposed to be associated with academic performance as they shape students' meaning systems, consolidating beliefs, goals, and actions in response to difficulties or setbacks [9]. For instance, students with growth mindsets view a test failure as a learning opportunity, sparking a learning-goal orientation to enhance their abilities (e.g., desire to take remedial learning after failing [10,11]), leading to practical actions or behaviors to achieve the goals [3]. Studies have found a positive association between growth mindsets and an extensive range of educational outcomes, including students' academic achievement [12–15], school engagement and motivation in learning [16,17], and students' well-being [18].

Given the discoveries of the association between mindsets and achievement, a wide range of education stakeholders, including parents, teachers, schools, training agencies, policymakers, and international organizations, such as the World Bank [19], actively encourage the implementation of growth mindset interventions to improve learners' achievement. Yet, several empirical studies controversially found null or weak effects of growth mindset on students' outcomes in specific cultural contexts. For instance, replication studies of mindset findings in Chinese populations have found that growth mindsets were either unrelated or only weakly related to expected academic outcomes, such as resilience and school grades [20,21]. Similarly, the OECD reported that the predictive effects of growth mindsets were limited among students in certain East Asian countries, based on findings from PISA 2018, which collected data from nationally representative samples of 15-year-old students in over 75 countries [22]. For the university level in the Czech Republic, one study argued that naturally holding growth mindsets did not predict test scores and the decision to retake the scholastic aptitude test, despite involving a large sample of over 5,500 university applicants [23].

Moreover, some studies discovered negative contributions of natural growth mindsets on academic achievement, suggesting that simply holding a growth mindset may not always be beneficial [24,25]. Similarly, several studies implementing growth mindset interventions failed to capture the change in academic achievement.

For example, a mindset training study with fourth-grade students from the United States found no significant difference in learning improvement between the intervention and non-intervention groups [26].

As the controversial discourse emerged, extensive meta-analytic reviews were conducted. Two meta-analyses found weak overall effects of naturally holding a growth mindset and mindset interventions on academic achievement [27]. Likewise, by reviewing 63 studies, another meta-analysis revealed faint overall effects of growth mindset interventions on academic achievement [28]. However, these studies reported potential meaningful benefits of growth mindsets for low socioeconomic or academically disadvantaged students, particularly in the context of growth mindset interventions. This closely aligns with the conclusion of a recent meta-analysis, which found that growth mindset interventions are more effective for at-risk groups, such as socioeconomically disadvantaged students, who are theoretically expected to benefit the most from adopting a growth mindset [7]. The heterogeneity captured by these meta-analyses supports increased attention to contextual moderators of the growth mindset-achievement link [29].

## Growth mindsets and socioeconomic status

One of the most common contextual moderators focused on by mindset advocates is socioeconomic status [30], as growth mindsets might be a promising approach for narrowing academic gaps between students from the top and bottom wealth quantiles [31,32]. A seminal nationwide study of Claro et al. [8] found that high school students in Chile from lower-income families who held growth mindsets achieved language and math scores comparable to those of their higher-income peers. Similarly, a large-scale study found that Latinx American students, who face social and economic challenges, showed improved grades after receiving a growth mindset intervention, resulting in the reduced achievement gaps with White students [33]. These findings raised the expectation that educational gaps from socioeconomic disparities may be bridged by promoting growth mindsets.

On the other hand, several studies reported distinct directions of the findings. According to PISA 2018 results, the benefits of growth mindsets were more promising among advantaged students than their disadvantaged peers in many Southeast Asian countries, with the exception of Singapore [22,34]. With the same cycle of PISA data, the associations between growth mindsets and achievement in math, science, and reading were significantly positive only among students with high socioeconomic status in the Philippines [30,35]. Additionally, a multilevel analysis, which nested students within schools and schools within countries and aggregated data from PISA 2018, concluded that growth mindsets were more beneficial for students from affluent families, schools, and countries [36]. These findings contrast with earlier evidence suggesting that growth mindsets may help narrow the achievement gap between socioeconomically advantaged and disadvantaged students.

In the United States, where numerous studies on growth mindsets have been conducted, the findings have been inconsistent. For instance, one study found that socioeconomic status moderated the association between growth mindsets and learning engagement among tenth-grade students, signifying that growth mindsets start working when students' socioeconomic status reaches certain levels [37]. On the other hand, another study focusing on ninth-grade students suggested that mindsets were equally beneficial for students regardless of their socioeconomic status [38]. Similarly, the results from PISA 2018 also showed that the achievement benefits of growth mindsets did not differ between students from higher and lower socioeconomic backgrounds in the United States [22].

The impact of growth mindsets on academic outcomes may not be uniform across societies, as it is shaped by the surrounding contexts in which students are immersed [39]. Recent empirical studies have argued that one's mindsets translate into actual behaviors and learning outcomes through a mechanism largely shaped by the social, cultural, and economic conditions of a particular environment. The social axioms, for instance, were found to moderate the relationship between growth mindsets and achievement [40]. The study discovered that the effect of growth mindsets was weaker in societies with stronger social complexity beliefs and in those where a problem is believed to have multiple solutions. Similarly, a study argued that socially shared values regarding growth mindsets themselves strengthen the influence of growth

mindsets on individual achievement [41]. At the same time, recent cross-national studies have demonstrated that national socioeconomic conditions, such as gross domestic product (GDP) and the indicators of educational mobility, could either amplify or suppress the magnitude to which growth mindsets are connected to academic growth [36,42]. These empirical pieces of evidence provide a theoretical basis for understanding the potential variation in the interaction between socio-economic status and growth mindset across countries, which underscores the need to interpret results within specific cultural contexts rather than assuming universal effects.

Taken together, the mixed findings across different studies raise concerns for policymakers and stakeholders in education, whether the benefits of growth mindsets can be generalized to their national or cultural contexts. Given these inconsistencies, applying growth mindset interventions without considering contextual factors may risk widening achievement gaps between disadvantaged and privileged students. Moreover, comparisons between studies and generalizations from them should be made with caution due to the heterogeneity in the methods used to measure mindsets and socioeconomic status, as well as the diversity of targeted educational outcomes.

## The present study

To address these inconsistencies, the present study aims to investigate the moderating role of socioeconomic status in the relationship between growth mindsets and math achievement across 74 different nations, using the cross-nationally standardized PISA 2022 dataset. To achieve this objective, we address two research questions. First, as a preliminary step, we ask: Does growth mindset predict math achievement? (*Research Question 1*). Second, as the primary focus of the study, we examine: Does socioeconomic status moderate the association between growth mindsets and math achievement? (*Research Question 2*).

We examine our research questions within each individual country rather than aggregating data across countries, as prior research has demonstrated that the benefits of growth mindsets can vary significantly by national context [22]. Aggregating results may obscure important country-specific patterns. Furthermore, country-level findings are more actionable and relevant for policymakers and stakeholders within specific national contexts.

To investigate these research questions across countries, we employed the most recent cycle of PISA, which contained data from 74 targeted countries. PISA datasets have been widely used in numerous studies, focusing on growth mindsets and academic achievement [43,44]. However, to our knowledge, no study has yet employed the PISA 2022 dataset to investigate the interplay between growth mindsets and socioeconomic status in predicting academic achievement across all PISA participating countries. Existing studies have typically focused on a single country [35] or a specific region [34], with one exception [36]; however, all have relied primarily on PISA 2018. The findings from PISA 2022 are expected to differ from those of previous cycles, given the widespread disruptions to education caused by the COVID-19 pandemic. Indeed, recent evidence indicates that the pandemic contributed to a decline in students' growth mindsets [45] and academic achievement [46].

Our study focuses on math achievement, given that it is universally recognized as an irreplaceable element for human cognitive and societal development. Math is one of the subjects that students find most challenging, although it facilitates metacognition, the competencies to plan, observe, and redress thoughts, while enhancing logical reasoning, abstraction, and pattern recognition [47]. Likewise, it always serves as a critical tool, driving advancement in science, technology, and economics across industries [48]. Therefore, math literacy is a vital asset not only for individual cognitive development and career opportunities [49], but also for countries' collective societal and economic advancement [50]. This profound importance is the rationale for this study's focus on math achievement. Additionally, math was the primary domain assessed in PISA 2022, making it more thoroughly measured than other subject areas [51].

It is also vital to acknowledge ongoing debates about what is actually being captured by the PISA's math test. PISA's test primarily focuses on cognitive skills such as the ability to solve problems and transfer knowledge to real-world situations [52], unlike curriculum-based assessments, such as TIMSS [53]. Recent studies have found that the PISA test

scores are associated with the general intelligence factor (g-factor) rather than subject-specific abilities [54,55]. In this study, math scores were employed as the outcome variable given their predictive power for future academic and occupational outcomes [56,57]. However, the findings should be interpreted with the understanding that PISA math scores may reflect both subject-specific knowledge and general cognitive abilities.

## Methodology

### Data source and sample

This study utilized data from the PISA 2022 assessment, which measured knowledge and skills from 15-year-old students in 81 countries and economies [51]. The de-identified dataset was retrieved from the PISA 2022 database at https://www.oecd.org/en/data/datasets/pisa-2022-database.html on April 17, 2025. We had no access to information that could identify individual participants during or after data collection. We downloaded the student questionnaire data file in SAV (SPSS) format, which included responses from students across all participating countries in the PISA 2022 dataset.

Our analysis focused on data from 74 countries for which growth mindsets, socioeconomic status indexes, and math scores were accessible. Due to the absence of data on growth mindsets, Cambodia, Guatemala, Israel, Paraguay, and Vietnam were excluded from the analysis. At the same time, Costa Rica was excluded because socioeconomic status data were unavailable. Additionally, Cyprus was not included in the study due to the unavailability of its data for public access [58]. Across the selected countries (total $N = 579{,}759$), the analytic sample consisted of 507,588 students, with 51.1% being girls. The number of samples per country ranged from 2,762–28,593 students (Table 1).

PISA 2022 targeted 15-year-old students enrolled in educational institutes, vocational education, and other related educational programs. The assessment employed a two-stage stratified sample design across all countries. In the first stage, schools were sampled from the list of eligible educational institutes in each country based on the probability proportional to the estimated number of 15-year-old students. Additionally, schools were grouped into strata to enhance the sampling accuracy. Then, in the second stage, approximately 40 students within the selected schools were sampled [58].

Given that this research utilized publicly available data, it was deemed to be exempt from ethics review by the Research Ethics Review Committee for Research Involving Human Research Participants, Group I, Chulalongkorn University, Thailand (COA No. 089/68) on April 11, 2025.

### Measures

**Growth mindset.** PISA employed a 4-point Likert scale, comprising four questions to assess students' perceptions of the malleability of certain attributes, specifically intelligence, math ability, language ability, and creativity. The respondents reported their level of agreement on a scale from 1 ("*strongly disagree*") to 4 ("*strongly agree*"). This study focused on the intelligence growth mindset, the belief that human intelligence is malleable. This mindset was measured using a single item (ST263Q02JA: "*Your intelligence is something about you that you cannot change very much.*"). This item was reversely coded so that higher scores reflected a stronger growth mindset.

The study decided to measure growth mindsets with a single item because this item had been well-validated by the development of the mindset scale [6] and utilized in several studies on mindsets [34,35]. This would enhance the compatibility when comparing the results of the current study with those of others. On the other hand, the PISA 2022 utilized a different format of questions to evaluate growth mindsets about intelligence and math. According to the OECD report [59], while the items on intelligence asked students about their own attributes, the items on math presented these attributes as being possessed by other people. The math growth mindset item asks: "*Some people are just not good at mathematics, no matter how hard they study.*" This item highly depended on how each respondent interpreted the term "some people." Given the potential for inconsistent connotations, this study used a growth mindset item related to intelligence, rather than one specific to math.

**Table 1. Descriptive information by country.**

| Country | Total sample | Analytic sample | PVMath (r1) | | Growth mind-sets (r2) | | SES (r3) | | Correlation | | |
|---|---|---|---|---|---|---|---|---|---|---|---|
| | | | Mean | SD | Mean | SD | Mean | SD | r1r2 | r1r3 | r2r3 |
| Austria | 6151 | 5671 | 494.9 | 90.7 | 2.93 | .89 | 0.09 | 0.93 | .11*** | .43*** | .05** |
| Estonia | 6392 | 6171 | 511.9 | 84.5 | 2.91 | .77 | 0.16 | 0.79 | .16*** | .37*** | .09*** |
| Germany | 6116 | 4967 | 489.2 | 92.1 | 2.90 | .83 | −0.09 | 1.03 | .06** | .42*** | .00 |
| Sweden | 6072 | 5526 | 490.9 | 91.6 | 2.88 | .82 | 0.36 | 0.83 | .07*** | .37*** | .07*** |
| Ireland | 5569 | 5431 | 493.8 | 78.5 | 2.88 | .73 | 0.34 | 0.80 | .28*** | .36*** | .13*** |
| United States | 4552 | 4050 | 469.4 | 94.7 | 2.86 | .84 | 0.07 | 0.98 | .28*** | .38*** | .12*** |
| Japan | 5760 | 5661 | 536.6 | 92.3 | 2.86 | .84 | −0.01 | 0.71 | .13*** | .35*** | .08*** |
| Kazakhstan | 19769 | 19179 | 426.5 | 78.4 | 2.82 | .80 | −0.37 | 0.83 | .09*** | .20*** | .05*** |
| New Zealand | 4682 | 4093 | 490.2 | 95.4 | 2.78 | .78 | 0.24 | 0.90 | .25*** | .40*** | .14*** |
| Australia | 13437 | 12580 | 491.9 | 97.8 | 2.77 | .78 | 0.39 | 0.84 | .29*** | .38*** | .15*** |
| Chile | 6488 | 5353 | 419.6 | 76.0 | 2.76 | .90 | −0.48 | 0.93 | .22*** | .36*** | .12*** |
| Brazil | 10798 | 8511 | 386.3 | 77.6 | 2.74 | .86 | −0.94 | 1.12 | .25*** | .39*** | .14*** |
| United Kingdom | 12972 | 10437 | 498.7 | 96.8 | 2.73 | .79 | 0.15 | 0.89 | .20*** | .34*** | .11*** |
| Canada | 23073 | 18923 | 505.5 | 92.6 | 2.72 | .82 | 0.40 | 0.74 | .18*** | .32*** | .09*** |
| Singapore | 6606 | 6536 | 575.6 | 102.6 | 2.72 | .87 | 0.31 | 0.83 | .15*** | .41*** | .08*** |
| Ukrainian regions[a] | 3876 | 3288 | 444.7 | 87.5 | 2.72 | .87 | −0.33 | 0.85 | .11*** | .37*** | .02 |
| Bulgaria | 6107 | 4798 | 429.6 | 95.8 | 2.71 | .93 | −0.22 | 1.03 | .05** | .43*** | .06*** |
| Denmark | 6200 | 5326 | 495.7 | 80.7 | 2.71 | .73 | 0.50 | 0.74 | .18*** | .34*** | .11*** |
| Switzerland | 6829 | 5932 | 517.1 | 93.4 | 2.70 | .85 | 0.20 | 0.92 | .04* | .45*** | .03* |
| Norway | 6611 | 5698 | 478.7 | 90.8 | 2.67 | .83 | 0.56 | 0.80 | .14*** | .30*** | .06*** |
| Latvia | 5373 | 4942 | 485.6 | 80.0 | 2.67 | .78 | 0.00 | 0.82 | .15*** | .36*** | .09*** |
| Georgia | 6583 | 4881 | 403.2 | 84.6 | 2.66 | .86 | −0.44 | 0.93 | .07*** | .30*** | .06** |
| Finland | 10239 | 8993 | 491.5 | 87.0 | 2.64 | .75 | 0.27 | 0.80 | .10*** | .35*** | .06*** |
| Iceland | 3360 | 2857 | 470.1 | 84.6 | 2.64 | .74 | 0.42 | 0.75 | .14*** | .29*** | .08*** |
| Lithuania | 7257 | 6692 | 479.3 | 86.2 | 2.63 | .74 | 0.06 | 0.88 | .07*** | .41*** | .08*** |
| Slovak Republic | 5824 | 5281 | 473.4 | 98.2 | 2.62 | .81 | −0.24 | 0.92 | .02 | .49*** | .04** |
| Hungary | 6198 | 5771 | 478.0 | 92.4 | 2.61 | .82 | 0.03 | 0.95 | .08*** | .50*** | .06*** |
| Uzbekistan | 7293 | 6309 | 369.2 | 66.9 | 2.61 | .96 | −0.67 | 1.00 | .03* | .14*** | .00 |
| Korea | 6454 | 6337 | 529.0 | 104.5 | 2.60 | .85 | 0.23 | 0.82 | .06** | .35*** | .00 |
| Chinese Taipei | 5857 | 5761 | 548.5 | 111.1 | 2.59 | .82 | −0.18 | 0.89 | .04 | .40*** | .00 |
| Uruguay | 6618 | 5262 | 421.6 | 81.0 | 2.59 | .82 | −0.76 | 1.13 | .13*** | .43*** | .08*** |
| Croatia | 6135 | 5791 | 467.1 | 86.7 | 2.58 | .80 | −0.14 | 0.83 | .05*** | .37*** | .03** |
| Morocco | 6867 | 5392 | 370.1 | 62.5 | 2.57 | .90 | −1.72 | 1.33 | −.02 | .26*** | 0.01 |
| Spain | 30800 | 28593 | 477.5 | 85.2 | 2.56 | .84 | −0.02 | 1.00 | .07*** | .37*** | .04*** |
| Portugal | 6793 | 6467 | 476.0 | 88.0 | 2.56 | .79 | −0.21 | 1.14 | .18*** | .42*** | .08*** |
| Poland | 6011 | 5544 | 495.5 | 87.0 | 2.56 | .83 | −0.09 | 0.88 | −.05** | .41*** | −.02 |
| Slovenia | 6721 | 6164 | 489.6 | 87.8 | 2.55 | .81 | 0.25 | 0.83 | .10*** | .39*** | .08*** |
| Argentina | 12111 | 9246 | 391.7 | 74.5 | 2.55 | .89 | −0.69 | 1.12 | .13*** | .40*** | .11*** |
| Jordan | 7799 | 6083 | 367.7 | 61.7 | 2.54 | .95 | −0.81 | 1.09 | −.02 | .25*** | .01 |
| Serbia | 6413 | 5850 | 445.9 | 87.9 | 2.54 | .86 | −0.18 | 0.83 | .03* | .36*** | .02 |
| United Arab Emirates | 24600 | 21009 | 440.6 | 101.5 | 2.54 | .93 | 0.31 | 0.74 | .20*** | .26*** | .08*** |
| Belgium | 8286 | 7132 | 500.2 | 91.4 | 2.54 | .79 | 0.13 | 0.90 | .04** | .44*** | .00 |
| Thailand | 8495 | 8229 | 395.8 | 75.4 | 2.53 | .77 | −1.23 | 1.13 | .14*** | .32*** | .07*** |
| Czech Republic | 8460 | 7849 | 492.2 | 92.0 | 2.52 | .77 | −0.07 | 0.85 | .02 | .45*** | .01 |

*(Continued)*

**Table 1.** (Continued)

| Country | Total sample | Analytic sample | PVMath (r1) | | Growth mind-sets (r2) | | SES (r3) | | Correlation | | |
|---|---|---|---|---|---|---|---|---|---|---|---|
| | | | Mean | SD | Mean | SD | Mean | SD | r1r2 | r1r3 | r2r3 |
| Malta | 3127 | 2767 | 474.3 | 96.3 | 2.52 | .88 | 0.04 | 0.96 | .16*** | .31*** | .08*** |
| Italy | 10552 | 10077 | 474.8 | 87.5 | 2.51 | .80 | −0.08 | 0.92 | .01 | .36*** | .00 |
| Mongolia | 6999 | 6514 | 429.3 | 82.4 | 2.51 | .95 | −0.70 | 1.06 | .07*** | .42*** | .02 |
| Montenegro | 5793 | 4992 | 413.6 | 81.1 | 2.50 | .91 | −0.18 | 0.84 | .00 | .31*** | .00 |
| Türkiye | 7250 | 7132 | 454.1 | 89.7 | 2.49 | .90 | −1.18 | 1.17 | .01 | .35*** | −.02* |
| Brunei Darussalam | 5576 | 5250 | 445.3 | 83.4 | 2.48 | .81 | −0.25 | 0.94 | .21*** | .40*** | .14*** |
| Peru | 6968 | 6167 | 399.7 | 76.5 | 2.47 | .79 | −1.07 | 1.22 | .19*** | .40*** | .12*** |
| Qatar | 7676 | 5762 | 426.9 | 89.3 | 2.46 | .92 | 0.14 | 0.84 | .11*** | .36*** | .07*** |
| Colombia | 7804 | 6792 | 390.6 | 72.3 | 2.45 | .82 | −1.02 | 1.20 | .17*** | .41*** | .09*** |
| Baku (Azerbaijan) | 7720 | 3932 | 418.1 | 85.7 | 2.45 | .94 | −0.48 | 0.93 | .00 | .24*** | .01 |
| Mexico | 6288 | 5975 | 398.4 | 68.3 | 2.44 | .81 | −0.92 | 1.16 | .13*** | .31*** | .06*** |
| France | 6770 | 5837 | 484.6 | 87.5 | 2.44 | .83 | 0.05 | 0.91 | .01 | .45*** | −.02 |
| Romania | 7364 | 6776 | 436.6 | 95.8 | 2.43 | .83 | −0.31 | 1.01 | .09*** | .50*** | .06*** |
| Malaysia | 7069 | 6646 | 411.4 | 75.7 | 2.40 | .76 | −0.66 | 1.04 | .03* | .43*** | .01 |
| Palestinian Authority | 7905 | 6503 | 370.8 | 66.2 | 2.40 | .92 | −0.90 | 1.06 | −.03* | .30*** | .03* |
| Netherlands | 5046 | 4499 | 506.1 | 101.8 | 2.39 | .76 | 0.30 | 0.84 | .06** | .36*** | .06** |
| Dominican Republic | 6868 | 4878 | 349.1 | 54.4 | 2.39 | .95 | −0.61 | 1.00 | .00 | .33*** | .01 |
| El Salvador | 6705 | 5438 | 351.0 | 59.5 | 2.37 | .85 | −1.33 | 1.23 | .05** | .40*** | .06** |
| Panama | 4544 | 3003 | 370.3 | 65.3 | 2.37 | .90 | −0.79 | 1.27 | .16*** | .44*** | .11*** |
| Saudi Arabia | 6928 | 6065 | 392.0 | 65.2 | 2.37 | .94 | −0.28 | 1.04 | .02 | .26*** | −.01 |
| Macao (China) | 4384 | 4355 | 552.2 | 92.4 | 2.37 | .78 | −0.45 | 0.91 | .02 | .22*** | .02 |
| Greece | 6403 | 6013 | 435.2 | 81.7 | 2.37 | .78 | −0.14 | 0.92 | −.02 | .34*** | −.02 |
| North Macedonia | 6610 | 5255 | 399.5 | 82.0 | 2.37 | .89 | −0.22 | 0.91 | .07*** | .35*** | .01 |
| Philippines | 7193 | 6829 | 357.0 | 64.8 | 2.36 | .75 | −1.33 | 1.13 | −.02 | .23*** | .07*** |
| Hong Kong (China) | 5907 | 5493 | 542.9 | 103.6 | 2.36 | .77 | −0.46 | 1.01 | −.02 | .24*** | .04* |
| Indonesia | 13439 | 12884 | 367.0 | 62.2 | 2.34 | .77 | −1.55 | 1.06 | .14*** | .24*** | .07*** |
| Republic of Moldova | 6235 | 5746 | 418.7 | 78.7 | 2.32 | .77 | −0.50 | 0.95 | −.07*** | .39*** | −.02 |
| Kosovo | 6027 | 4779 | 361.7 | 62.9 | 2.28 | .92 | −0.32 | 0.88 | −.01 | .25*** | −.01 |
| Jamaica | 3873 | 2762 | 391.2 | 71.4 | 2.27 | .92 | −0.48 | 0.93 | .21*** | .26*** | .10*** |
| Albania | 6129 | 3901 | 379.0 | 86.8 | 2.22 | .93 | −0.77 | 1.09 | −.03 | .24*** | .01 |

Note: All statistics were pooled across 10 plausible values and weighted using W_FSTUWT.

*p < .05, **p < .01, ***p < .001.

The countries were sorted in descending order of the mean of growth mindsets.

ªThe results refer only to 18 out of 27 regions.

**Socioeconomic status.** This study used PISA's economic, social, and cultural (ESCS) index as a proxy for students' socioeconomic status. The variable was computed based on three key indicators: parental occupation status, parental educational attainment, and domestic possession [58]. The variable was standardized to have a mean of 0 and a standard deviation of 1 across OECD countries. Higher scores indicate higher socioeconomic status, representing advantaged students.

In educational and social science research, socioeconomic status is typically constituted upon observable indicators such as parental education, parental occupation, and household wealth and income [60]. These dimensions

were incorporated in PISA to assess socioeconomic status; however, several studies have highlighted limitations of the ESCS in PISA, proposing issues of comparability across different education systems and the trustworthiness of student self-reports, which hindered the explanatory power of the socioeconomic status on achievement [61,62]. Moreover, discrepancies of math achievement between student self-reported socioeconomic status in the international large-scale assessment and national socioeconomic status measures were observed by a recent study [63]. This suggested that the assumed proxies of socioeconomic status may underestimate the influence of related structural disadvantage. Nevertheless, in contrast, one study did not observe different conclusions from using different indicators of socioeconomic status, namely family wealth (the availability of household items in the country context) and home possession (number of items their household possessed in a broader range), to predict the association between students' growth mindsets and math achievement in the Philippines using PISA 2018 [30]. Although we acknowledge that the ESCS does not fully capture structural and multidimensional aspects of socioeconomic conditions, it provides a standardized baseline that enables cross-cultural analysis, which is central to the aims of this study.

**Math achievement.**  The PISA 2022 assessment measured students' math literacy using plausible values, which are a set of ten test scores (PV1MATH–PV10MATH) that a student could possibly achieve. As students were randomly assigned to various test questions, these plausible values were utilized to mitigate potential errors. In our analysis, we included all plausible values in accordance with the guidelines outlined in the PISA Data Analysis Manual [64]. Each plausible value variable was standardized to have a mean of 500 and a standard deviation of 100 across OECD countries.

## Data analysis

To address our research questions, we performed regression and moderation analyses individually across countries, using Stata version 18. We employed the Stata macro "repest," developed by OECD, to account for the complex survey design and the analysis of plausible values in PISA [65]. The macro incorporated 10 plausible values of math scores, along with the appropriate weight variables, following the analytical procedures recommended by the OECD [64]. PISA survey design involved two types of weights: final weights and replicate weights. The final weights variable (W_FSTUWT) and 80 replicates (W_FSTURWT1–W_FSTURWT80) were used to calculate unbiased estimates of population parameters and standard error estimates, respectively. The cleaned dataset used in this study and the Stata scripts presenting steps of the main analyses are available on the Open Science Framework platform at https://osf.io/b2a6k/?view_only=9a352016b926445f8130b5ad18eb9d56.

Regression analyses were conducted for each country to address Research Question 1: *Does growth mindset predict math achievement?* In these regression models, math achievement was regressed on growth mindset and socioeconomic status without including any interaction terms. To address Research Question 2: *Does socioeconomic status moderate the association between growth mindset and math achievement?* We conducted moderation analyses to examine whether the strength of the association between growth mindset and math achievement varied across levels of socioeconomic status. This involved adding an interaction term between growth mindset and socioeconomic status to each country's regression model.

To probe the interaction, the predicted math achievement scores were computed at two points of growth mindset and socioeconomic levels based on the "pick-a-point" approach. These points corresponded to the 16th and 84th percentiles, representing low and high levels for each construct [66]. Following this, the score-point difference in math achievement between students with low and high growth mindsets was calculated. This difference reflected the performance gains associated with adopting growth mindsets compared to fixed mindsets. Finally, we plotted the gap in math score differences between students from low- and high-socioeconomic backgrounds to demonstrate whether growth mindsets may provide more benefits for those facing economic challenges.

## Results

Descriptive statistics, bivariate correlation, mean, and standard deviation, pooled across 10 datasets for plausible values, by country, are reported in Table 1. Across countries, correlations between math scores and growth mindsets were generally modest (−.07 to.29), those between math scores and socioeconomic status were stronger (.14 to.50), and those between growth mindsets and socioeconomic status were weaker and often negligible (−.02 to.15).

### Main analysis

To answer the first research question, we present the results of regression models (Model 1), focusing on the predictive effect of growth mindsets on math achievement after controlling for socioeconomic status. The detailed regression results by country are presented in Table 2.

The results showed that growth mindsets were significantly positively associated with math achievement ($p < .05$) in 50 out of 74 countries (67.6%). The unstandardized regression coefficients for growth mindsets ranged widely, from 2.22 to 30.26, while the standardized estimates were between −0.05 and 0.25. The top five countries exhibiting the largest positive coefficients are as follows: Australia, the United States, Ireland, New Zealand, and the United Kingdom, respectively.

In contrast, holding a growth mindset was significantly associated with lower math achievement in the Palestinian Authority, the Philippines, Hong Kong (China), Poland, and the Republic of Moldova. In these countries, students with fixed mindsets outperformed their peers who held growth mindsets, with unstandardized coefficients ranging from −3.01 (β = −.03) to −6.45 (β = −.05).

In addition, the predictive effect of growth mindsets was not statistically significant in 19 out of 74 countries (25.7%). Fig 1 presents these countries in a lighter tone. The unstandardized coefficients of growth mindsets ranged from −3.07 (β = −.03) to 2.94 (β = .02). The nonsignificant group included a range of countries across various regions, including the Balkans (e.g., Albania, Bulgaria, and Kosovo), the Turkic region (e.g., Türkiye, Azerbaijan, and Uzbekistan), the Western Alps (e.g., France, Italy, and Switzerland), the Middle East (e.g., Jordan and Saudi Arabia), and other nations (e.g., Malaysia, El Salvador, and Macao (China)). In sum, the association between growth mindsets and math achievement was not uniform. It varied across national contexts.

Regarding the second research question, the results showed that socioeconomic status significantly moderated the association between growth mindsets and math achievement in 33 out of 74 countries (44.6%), indicating that the predictive effect of growth mindsets on math achievement varied by students' socioeconomic status. In contrast, no significant moderation was found in 41 countries (55.4%), suggesting that the association between growth mindsets and achievement was similar across socioeconomic levels in those countries.

According to the moderation results (Model 2) for all 74 countries reported in Table 3, the estimates of growth mindsets ranged from −4.26 (β = −.03) and 29.36 (β = .24) score points, while those of SES were between −1.07 (β = .36) and 66.05 (β = .56). The estimated interaction terms also observed a wide range, having the unstandardized coefficients between −6.01 (β = −.06) to 13.89 (β = .13) points. The R-squared estimates ranged from.02 to.25. In addition, the moderation plots showing the association between growth mindsets and math achievement at high and low socioeconomic status levels across all countries are presented in S1 Appendix.

Among the 33 countries that showed significant moderation, seven countries (e.g., Singapore, Croatia, and Serbia) had a significantly *negative* interaction between growth mindsets and socioeconomic status, with unstandardized coefficients ranging from −2.77 to −6.01. The standardized estimates were between −0.03 and −0.06. This suggested that in about one-tenth of the countries, higher socioeconomic status was associated with a smaller association between growth mindsets and math achievement.

In contrast, in 26 countries (e.g., Qatar, United Arab Emirates, and Malaysia), the interaction term was significant in a *positive* direction, with unstandardized estimates ranging from 2.26 (β = 0.02) to 13.89 (β = 0.13). This suggested that

**Table 2. Multiple regression models predicting math achievement by country (Model 1).**

| Country | Growth mindset | | | | SES | | | | Constants | R² |
|---|---|---|---|---|---|---|---|---|---|---|
| | β | B | 95% CI | | β | B | 95% CI | | | |
| | | | Lower | Upper | | | Lower | Upper | | |
| Republic of Moldova | −.05 | **−6.45** | −9.37 | −3.53 | .36 | **32.35** | 29.14 | 35.56 | 449.8 | .16 |
| Poland | −.04 | **−4.22** | −7.08 | −1.37 | .44 | **39.89** | 36.12 | 43.66 | 510.0 | .17 |
| Hong Kong (China) | −.03 | **−3.98** | −7.90 | −0.07 | .28 | **24.91** | 20.46 | 29.36 | 563.8 | .06 |
| Philippines | −.03 | **−3.47** | −6.25 | −0.70 | .15 | **13.53** | 9.92 | 17.13 | 383.3 | .06 |
| Albania | −.03 | −3.07 | −6.71 | 0.56 | .21 | **19.12** | 15.22 | 23.02 | 400.5 | .06 |
| Palestinian Authority | −.03 | **−3.01** | −4.94 | −1.09 | .21 | **18.87** | 16.44 | 21.31 | 395.0 | .09 |
| Morocco | −.01 | −1.37 | −3.60 | 0.86 | .13 | **12.15** | 8.85 | 15.45 | 394.5 | .07 |
| Greece | −.01 | −1.15 | −4.46 | 2.16 | .34 | **30.70** | 27.74 | 33.66 | 442.1 | .12 |
| Jordan | −.01 | −1.13 | −3.27 | 1.00 | .16 | **14.16** | 11.44 | 16.87 | 382.1 | .06 |
| Slovak Republic | −.01 | −0.71 | −3.97 | 2.56 | .58 | **52.22** | 47.41 | 57.03 | 487.5 | .24 |
| Kosovo | −.01 | −0.65 | −2.94 | 1.64 | .20 | **18.03** | 15.40 | 20.66 | 368.9 | .06 |
| Baku (Azerbaijan) | .00 | −0.29 | −3.51 | 2.94 | .25 | **22.56** | 18.18 | 26.95 | 429.6 | .06 |
| Montenegro | .00 | −0.12 | −2.96 | 2.71 | .33 | **29.60** | 26.40 | 32.79 | 419.2 | .09 |
| Dominican Republic | .00 | 0.05 | −1.90 | 2.01 | .20 | **17.93** | 15.14 | 20.71 | 360.0 | .11 |
| Italy | .01 | 0.97 | −1.92 | 3.86 | .38 | **34.33** | 29.99 | 38.67 | 475.3 | .13 |
| Saudi Arabia | .02 | 1.82 | −0.39 | 4.03 | .18 | **16.52** | 13.98 | 19.05 | 392.4 | .07 |
| France | .02 | 1.90 | −0.94 | 4.74 | .49 | **43.73** | 40.63 | 46.83 | 477.9 | .21 |
| Türkiye | .02 | 2.11 | −0.64 | 4.85 | .30 | **27.17** | 24.67 | 29.67 | 481.0 | .13 |
| El Salvador | .02 | 2.16 | −0.28 | 4.59 | .21 | **19.25** | 16.67 | 21.82 | 371.5 | .16 |
| Macao (China) | .02 | 2.22 | −1.11 | 5.54 | .25 | **22.80** | 19.70 | 25.90 | 557.2 | .05 |
| Uzbekistan | .02 | **2.22** | 0.32 | 4.12 | .11 | **9.61** | 7.04 | 12.18 | 369.8 | .02 |
| Czech Republic | .02 | 2.22 | −0.52 | 4.96 | .54 | **48.64** | 45.13 | 52.15 | 490.1 | .20 |
| Malaysia | .02 | 2.24 | −1.70 | 6.19 | .34 | **31.03** | 26.91 | 35.15 | 426.7 | .18 |
| Bulgaria | .02 | 2.46 | −0.78 | 5.69 | .44 | **39.63** | 34.93 | 44.33 | 431.6 | .18 |
| Switzerland | .02 | 2.94 | −0.08 | 5.95 | .51 | **46.01** | 42.82 | 49.19 | 499.9 | .21 |
| Serbia | .02 | **2.96** | 0.10 | 5.83 | .42 | **37.81** | 31.52 | 44.10 | 445.3 | .13 |
| Croatia | .04 | **4.21** | 1.16 | 7.27 | .43 | **38.61** | 34.71 | 42.50 | 461.8 | .14 |
| Lithuania | .04 | **4.46** | 1.80 | 7.12 | .44 | **39.80** | 36.54 | 43.06 | 465.1 | .17 |
| Belgium | .04 | **4.84** | 1.73 | 7.95 | .50 | **44.88** | 41.94 | 47.82 | 481.9 | .20 |
| Chinese Taipei | .04 | **4.96** | 0.49 | 9.43 | .55 | **49.91** | 43.95 | 55.88 | 544.8 | .16 |
| Netherlands | .04 | **5.08** | 0.04 | 10.11 | .48 | **43.40** | 38.57 | 48.24 | 480.8 | .13 |
| Mongolia | .04 | **5.12** | 2.75 | 7.50 | .36 | **32.64** | 29.46 | 35.83 | 439.2 | .18 |
| Sweden | .05 | **5.40** | 1.70 | 9.10 | .45 | **40.55** | 37.21 | 43.88 | 460.8 | .14 |
| Georgia | .05 | **5.67** | 2.25 | 9.10 | .30 | **27.00** | 22.29 | 31.70 | 399.9 | .09 |
| Hungary | .05 | **5.91** | 3.26 | 8.55 | .53 | **48.20** | 44.61 | 51.78 | 461.3 | .25 |
| Spain | .05 | **6.14** | 4.11 | 8.17 | .34 | **31.03** | 29.32 | 32.73 | 462.3 | .14 |
| North Macedonia | .05 | **6.40** | 3.56 | 9.23 | .34 | **30.91** | 28.32 | 33.50 | 391.2 | .12 |
| Romania | .06 | **6.69** | 3.83 | 9.55 | .52 | **46.65** | 42.53 | 50.77 | 434.8 | .25 |
| Germany | .06 | **6.72** | 3.23 | 10.20 | .42 | **38.08** | 35.11 | 41.06 | 473.3 | .18 |
| Korea | .06 | **7.05** | 1.86 | 12.24 | .50 | **45.00** | 39.08 | 50.91 | 500.4 | .13 |
| Kazakhstan | .06 | **7.62** | 5.75 | 9.49 | .20 | **18.36** | 15.77 | 20.94 | 411.8 | .05 |
| Argentina | .06 | **7.75** | 4.89 | 10.61 | .28 | **25.52** | 22.99 | 28.05 | 389.5 | .16 |
| Slovenia | .07 | **8.04** | 4.33 | 11.76 | .45 | **40.28** | 37.20 | 43.36 | 459.1 | .16 |
| Panama | .07 | **8.26** | 4.97 | 11.56 | .24 | **21.80** | 18.06 | 25.53 | 367.8 | .20 |

*(Continued)*

**Table 2.** (Continued)

| Country | Growth mindset | | | | SES | | | | Constants | $R^2$ |
|---|---|---|---|---|---|---|---|---|---|---|
| | β | B | 95% CI | | β | B | 95% CI | | | |
| | | | Lower | Upper | | | Lower | Upper | | |
| Qatar | .07 | **8.83** | 5.99 | 11.67 | .42 | **38.02** | 34.81 | 41.24 | 400.0 | .14 |
| Austria | .08 | **9.09** | 6.38 | 11.79 | .46 | **41.85** | 39.07 | 44.64 | 464.4 | .20 |
| Finland | .08 | **9.34** | 6.54 | 12.14 | .42 | **37.63** | 34.75 | 40.51 | 456.6 | .13 |
| Mexico | .08 | **9.43** | 6.84 | 12.02 | .20 | **18.16** | 15.56 | 20.77 | 392.0 | .11 |
| Indonesia | .08 | **9.83** | 7.27 | 12.40 | .15 | **13.78** | 10.51 | 17.06 | 365.4 | .07 |
| Uruguay | .08 | **9.85** | 6.57 | 13.13 | .33 | **30.01** | 27.42 | 32.60 | 418.9 | .19 |
| Ukrainian regions[a] | .08 | **10.09** | 5.12 | 15.05 | .42 | **37.93** | 30.87 | 44.99 | 429.9 | .15 |
| Japan | .09 | **10.77** | 7.52 | 14.02 | .49 | **43.75** | 37.76 | 49.74 | 506.1 | .13 |
| Thailand | .09 | **11.21** | 8.21 | 14.20 | .23 | **21.11** | 16.82 | 25.40 | 393.4 | .12 |
| Colombia | .10 | **11.61** | 9.06 | 14.16 | .26 | **23.79** | 20.41 | 27.17 | 386.3 | .18 |
| Latvia | .10 | **12.06** | 9.16 | 14.96 | .37 | **33.79** | 30.67 | 36.91 | 453.4 | .14 |
| Norway | .11 | **13.13** | 9.72 | 16.53 | .37 | **33.63** | 30.15 | 37.11 | 425.0 | .11 |
| Iceland | .11 | **13.42** | 8.55 | 18.30 | .35 | **31.15** | 26.74 | 35.56 | 421.4 | .10 |
| Singapore | .11 | **13.49** | 10.59 | 16.39 | .55 | **49.92** | 46.62 | 53.22 | 523.6 | .18 |
| Peru | .12 | **13.93** | 11.13 | 16.72 | .27 | **24.33** | 22.24 | 26.41 | 391.3 | .18 |
| Estonia | .12 | **14.05** | 10.77 | 17.33 | .42 | **37.95** | 34.53 | 41.37 | 465.1 | .15 |
| Jamaica | .12 | **14.41** | 10.55 | 18.27 | .20 | **18.45** | 15.04 | 21.86 | 367.4 | .10 |
| Malta | .12 | **14.49** | 9.82 | 19.16 | .33 | **29.57** | 26.17 | 32.97 | 436.6 | .11 |
| Chile | .13 | **15.27** | 12.49 | 18.06 | .31 | **27.74** | 24.91 | 30.57 | 390.8 | .16 |
| Portugal | .13 | **15.98** | 13.02 | 18.94 | .35 | **31.96** | 29.51 | 34.41 | 441.9 | .20 |
| Denmark | .13 | **16.10** | 12.31 | 19.89 | .40 | **35.67** | 32.32 | 39.01 | 434.2 | .14 |
| Brunei Darussalam | .14 | **16.34** | 13.55 | 19.14 | .37 | **33.43** | 31.40 | 35.46 | 412.9 | .18 |
| Canada | .14 | **17.15** | 14.71 | 19.58 | .43 | **38.33** | 35.04 | 41.61 | 443.6 | .13 |
| Brazil | .15 | **17.84** | 15.64 | 20.03 | .28 | **25.02** | 22.53 | 27.52 | 361.0 | .19 |
| United Arab Emirates | .16 | **19.42** | 17.87 | 20.97 | .38 | **33.81** | 31.35 | 36.27 | 380.9 | .10 |
| United Kingdom | .17 | **20.27** | 16.88 | 23.67 | .38 | **34.60** | 29.59 | 39.61 | 438.2 | .14 |
| New Zealand | .20 | **23.85** | 20.12 | 27.58 | .44 | **39.21** | 35.34 | 43.09 | 414.3 | .20 |
| Ireland | .21 | **25.38** | 22.86 | 27.89 | .35 | **31.98** | 29.03 | 34.93 | 409.8 | .18 |
| United States | .23 | **26.95** | 23.14 | 30.75 | .38 | **34.53** | 29.91 | 39.14 | 390.0 | .20 |
| Australia | .25 | **30.26** | 27.79 | 32.74 | .45 | **40.40** | 37.71 | 43.10 | 392.3 | .20 |

Note: All statistics were pooled across 10 plausible values and weighted using W_FSTUWT.

β represents standardized coefficients, while B refers to unstandardized coefficients.

Statistically significant coefficients (p < .05) were presented in bold.

The countries were sorted in ascending order of the growth mindset's coefficient size.

[a]The results refer only to 18 out of 27 regions.

students with higher socioeconomic status benefited more from holding growth mindsets in approximately one-third of the countries.

To visually illustrate the benefits of growth mindsets, the math score-point difference between students with growth mindsets and those with fixed mindsets was plotted for each country, as shown in Fig 2. Black-filled bars represent the score difference among students with low socioeconomic status, while non-filled bars illustrate that of students with high socioeconomic status. Countries on the Y-axis are sorted in ascending order based on the mindset achievement gaps

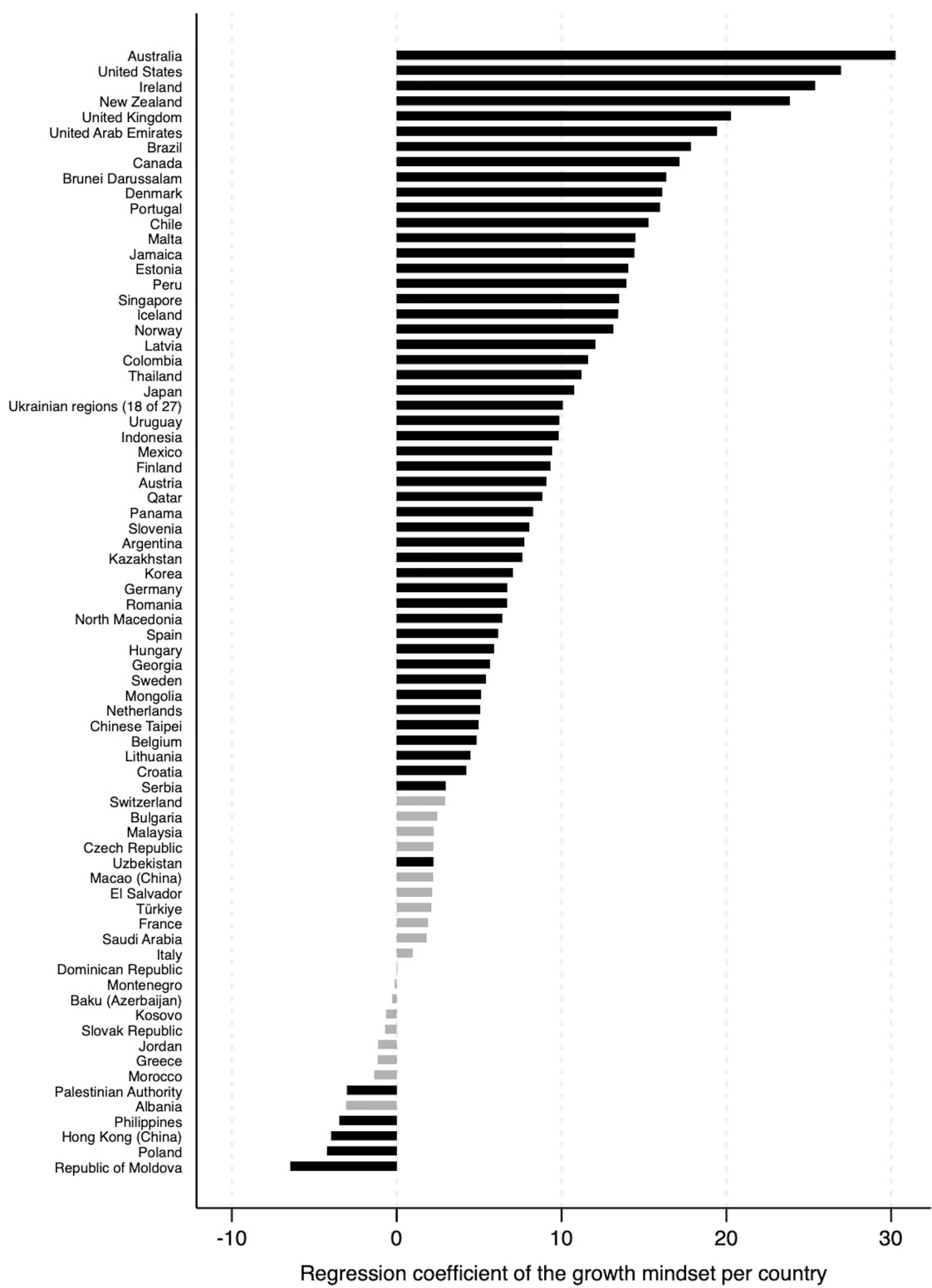

**Fig 1. The coefficients of growth mindsets in predicting math achievement.** Note: Darker-toned bars illustrate that the effect of growth mindsets was statistically significant (p<.05).

**Table 3. Results from the moderation models (Model 2) predicting math achievement by country.**

| Country | Growth mindsets | | | | SES | | | | Growth mindsets x SES | | | | Constant | R² |
|---|---|---|---|---|---|---|---|---|---|---|---|---|---|---|
| | β | B | 95% CI | | β | B | 95% CI | | β | B | 95% CI | | | |
| | | | Lower | Upper | | | Lower | Upper | | | Lower | Upper | | |
| Singapore | .15 | **15.52** | 11.97 | 19.06 | .56 | **66.05** | 55.86 | 76.23 | −.06 | **−6.01** | −9.54 | −2.48 | 518.5 | .19 |
| Croatia | .04 | **3.69** | 0.58 | 6.80 | .43 | **50.83** | 41.89 | 59.76 | −.04 | **−4.76** | −8.08 | −1.44 | 463.2 | .14 |
| Serbia | .03 | 2.25 | −0.67 | 5.17 | .42 | **49.16** | 36.14 | 62.19 | −.04 | **−4.51** | −8.62 | −0.40 | 447.2 | .13 |
| Austria | .09 | **9.42** | 6.73 | 12.11 | .48 | **54.00** | 44.78 | 63.21 | −.04 | **−4.12** | −7.06 | −1.19 | 463.6 | .20 |
| Germany | .06 | **6.45** | 2.97 | 9.93 | .44 | **48.80** | 38.90 | 58.70 | −.04 | **−3.71** | −6.83 | −0.58 | 474.1 | .18 |
| Italy | .01 | 0.76 | −2.07 | 3.60 | .38 | **42.39** | 34.54 | 50.24 | −.03 | **−3.20** | −5.64 | −0.77 | 475.8 | .13 |
| Chinese Taipei | .05 | **4.52** | 0.10 | 8.94 | .55 | **57.88** | 43.21 | 72.56 | −.03 | −3.08 | −7.42 | 1.26 | 545.9 | .16 |
| Türkiye | .00 | −1.22 | −5.24 | 2.81 | .30 | **34.05** | 28.08 | 40.01 | −.03 | **−2.77** | −4.84 | −0.70 | 489.2 | .13 |
| Ireland | .22 | **26.14** | 23.36 | 28.92 | .36 | **38.67** | 28.12 | 49.22 | −.02 | −2.35 | −5.76 | 1.07 | 407.8 | .18 |
| United Kingdom | .18 | **20.49** | 16.97 | 24.01 | .39 | **40.42** | 29.25 | 51.60 | −.02 | −2.15 | −6.09 | 1.79 | 437.8 | .14 |
| Denmark | .15 | **17.13** | 13.09 | 21.17 | .40 | **40.78** | 29.69 | 51.86 | −.02 | −1.94 | −5.89 | 2.01 | 431.6 | .14 |
| Estonia | .12 | **14.35** | 10.95 | 17.76 | .43 | **43.10** | 31.83 | 54.37 | −.02 | −1.79 | −5.67 | 2.09 | 464.3 | .15 |
| Montenegro | .00 | −0.36 | −3.20 | 2.47 | .33 | **33.94** | 25.79 | 42.09 | −.02 | −1.74 | −4.71 | 1.22 | 419.8 | .10 |
| Spain | .05 | **6.05** | 4.02 | 8.08 | .34 | **34.83** | 30.91 | 38.74 | −.01 | −1.48 | −2.99 | 0.02 | 462.6 | .14 |
| France | .02 | 1.96 | −0.88 | 4.79 | .48 | **47.01** | 39.81 | 54.20 | −.01 | −1.34 | −3.89 | 1.22 | 477.7 | .21 |
| Portugal | .14 | **15.76** | 12.74 | 18.79 | .35 | **35.25** | 28.64 | 41.85 | −.01 | −1.30 | −3.66 | 1.06 | 442.5 | .20 |
| Switzerland | .03 | 3.12 | −0.02 | 6.27 | .51 | **48.46** | 38.02 | 58.90 | −.01 | −0.92 | −4.39 | 2.56 | 499.4 | .21 |
| Japan | .09 | **10.76** | 7.51 | 14.01 | .49 | **46.10** | 33.66 | 58.53 | −.01 | −0.82 | −5.07 | 3.43 | 506.2 | .13 |
| Poland | −.03 | **−4.26** | −7.15 | −1.37 | .44 | **41.24** | 31.27 | 51.21 | −.01 | −0.53 | −4.13 | 3.07 | 510.1 | .17 |
| Czech Republic | .02 | 2.20 | −0.57 | 4.97 | .54 | **49.40** | 38.80 | 60.00 | .00 | −0.31 | −3.99 | 3.37 | 490.1 | .20 |
| Albania | −.03 | −3.27 | −7.87 | 1.32 | .21 | **19.76** | 12.43 | 27.09 | .00 | −0.29 | −3.60 | 3.03 | 401.0 | .06 |
| Sweden | .05 | **5.47** | 1.32 | 9.62 | .45 | **41.04** | 27.36 | 54.73 | .00 | −0.17 | −4.62 | 4.27 | 460.6 | .14 |
| Lithuania | .04 | **4.47** | 1.83 | 7.11 | .44 | **39.93** | 29.92 | 49.94 | .00 | −0.05 | −3.73 | 3.63 | 465.1 | .17 |
| Belgium | .04 | **4.84** | 1.71 | 7.97 | .50 | **44.91** | 36.91 | 52.92 | .00 | −0.01 | −3.01 | 2.98 | 481.9 | .20 |
| Hungary | .05 | **5.88** | 3.23 | 8.53 | .53 | **46.44** | 38.26 | 54.63 | .01 | 0.68 | −2.18 | 3.54 | 461.3 | .25 |
| Jordan | −.01 | −0.53 | −3.40 | 2.33 | .16 | **12.26** | 6.98 | 17.54 | .01 | 0.73 | −1.18 | 2.65 | 380.5 | .06 |
| Finland | .07 | **9.10** | 6.14 | 12.05 | .42 | **35.37** | 25.43 | 45.32 | .01 | 0.84 | −2.60 | 4.28 | 457.2 | .13 |
| Uzbekistan | .02 | **2.85** | 0.49 | 5.21 | .11 | **7.13** | 1.57 | 12.69 | .01 | 0.95 | −0.95 | 2.86 | 368.2 | .02 |
| Korea | .05 | **6.77** | 1.23 | 12.32 | .50 | **42.18** | 23.06 | 61.31 | .01 | 1.08 | −5.62 | 7.78 | 501.1 | .13 |
| Greece | −.01 | −1.01 | −4.33 | 2.32 | .34 | **28.06** | 20.08 | 36.04 | .01 | 1.11 | −1.84 | 4.06 | 441.8 | .12 |
| Morocco | .00 | 0.82 | −3.30 | 4.94 | .13 | **8.78** | 3.59 | 13.96 | .01 | 1.29 | −0.40 | 2.97 | 388.8 | .07 |
| Netherlands | .04 | 4.70 | −0.34 | 9.74 | .48 | **40.26** | 30.02 | 50.51 | .01 | 1.32 | −2.66 | 5.31 | 481.6 | .13 |
| Chile | .13 | **15.95** | 13.04 | 18.87 | .31 | **23.89** | 16.10 | 31.67 | .01 | 1.41 | −1.21 | 4.02 | 388.8 | .16 |
| Mongolia | .05 | **6.10** | 3.27 | 8.92 | .36 | **29.05** | 22.78 | 35.33 | .01 | 1.42 | −0.78 | 3.61 | 436.7 | .18 |
| Republic of Moldova | −.05 | **−5.81** | −9.21 | −2.41 | .36 | **29.03** | 21.02 | 37.04 | .01 | 1.43 | −1.75 | 4.61 | 448.3 | .16 |
| United States | .22 | **26.89** | 23.10 | 30.67 | .38 | **30.37** | 18.36 | 42.38 | .01 | 1.46 | −2.23 | 5.16 | 390.1 | .20 |
| Macao (China) | .02 | 2.84 | −0.76 | 6.43 | .26 | **19.29** | 10.59 | 27.98 | .01 | 1.47 | −2.17 | 5.10 | 555.7 | .05 |
| Malta | .12 | **14.31** | 9.73 | 18.89 | .33 | **25.46** | 13.63 | 37.29 | .02 | 1.63 | −2.97 | 6.23 | 437.0 | .11 |
| Saudi Arabia | .01 | 2.27 | −0.11 | 4.66 | .19 | **12.60** | 7.58 | 17.61 | .02 | 1.63 | −0.42 | 3.68 | 391.3 | .07 |
| Palestinian Authority | −.02 | −1.62 | −4.40 | 1.15 | .21 | **14.94** | 9.69 | 20.19 | .02 | 1.64 | −0.43 | 3.72 | 391.6 | .09 |
| Ukrainian regions | .08 | **10.61** | 5.46 | 15.76 | .42 | **32.72** | 17.68 | 47.76 | .02 | 1.86 | −4.56 | 8.28 | 428.4 | .15 |
| Slovak Republic | −.01 | −0.24 | −3.43 | 2.95 | .58 | **47.18** | 37.27 | 57.10 | .02 | 1.92 | −1.84 | 5.67 | 486.3 | .24 |
| Canada | .13 | **16.36** | 13.88 | 18.85 | .42 | **32.85** | 23.98 | 41.72 | .02 | 2.03 | −0.73 | 4.78 | 445.6 | .13 |
| Dominican Republic | .01 | 1.48 | −0.80 | 3.76 | .20 | **12.50** | 7.74 | 17.25 | .02 | **2.26** | 0.64 | 3.89 | 356.5 | .11 |

*(Continued)*

**Table 3.** (Continued)

| Country | Growth mindsets | | | | SES | | | | Growth mindsets x SES | | | | Constant | R² |
|---|---|---|---|---|---|---|---|---|---|---|---|---|---|---|
| | β | B | 95% CI | | β | B | 95% CI | | β | B | 95% CI | | | |
| | | | Lower | Upper | | | Lower | Upper | | | Lower | Upper | | |
| Argentina | .07 | **9.33** | 6.07 | 12.59 | .28 | **19.73** | 14.21 | 25.25 | .02 | **2.27** | 0.23 | 4.31 | 385.2 | .17 |
| Uruguay | .09 | **11.72** | 8.14 | 15.31 | .33 | **23.76** | 17.12 | 30.39 | .02 | **2.40** | 0.07 | 4.72 | 413.9 | .19 |
| Australia | .24 | **29.36** | 26.61 | 32.10 | .44 | **33.90** | 26.07 | 41.74 | .02 | 2.40 | −0.39 | 5.19 | 394.6 | .20 |
| Kosovo | −.01 | 0.12 | −2.26 | 2.51 | .21 | **12.51** | 6.93 | 18.09 | .02 | **2.43** | 0.04 | 4.81 | 367.2 | .07 |
| Baku (Azerbaijan) | .00 | 1.02 | −2.98 | 5.03 | .25 | **15.76** | 5.87 | 25.66 | .03 | 2.79 | −1.17 | 6.75 | 426.4 | .06 |
| New Zealand | .19 | **23.24** | 19.35 | 27.13 | .43 | **31.34** | 19.78 | 42.91 | .03 | 2.88 | −1.04 | 6.79 | 415.7 | .20 |
| Slovenia | .05 | **7.22** | 3.39 | 11.04 | .45 | **32.99** | 22.38 | 43.59 | .03 | 2.89 | −1.31 | 7.09 | 461.1 | .16 |
| Hong Kong (China) | −.03 | −2.55 | −6.85 | 1.74 | .28 | **17.20** | 7.94 | 26.47 | .03 | 3.24 | −0.61 | 7.09 | 560.3 | .06 |
| Mexico | .10 | **12.61** | 9.72 | 15.50 | .21 | **9.10** | 4.91 | 13.30 | .03 | **3.66** | 2.01 | 5.31 | 384.0 | .11 |
| Bulgaria | .02 | 3.34 | −0.08 | 6.76 | .44 | **29.53** | 19.09 | 39.97 | .04 | **3.75** | 0.31 | 7.19 | 429.0 | .19 |
| Philippines | .00 | 0.89 | −3.32 | 5.11 | .16 | 4.47 | −0.13 | 9.06 | .04 | **3.76** | 1.74 | 5.78 | 372.5 | .06 |
| Colombia | .12 | **15.47** | 12.54 | 18.41 | .27 | **13.54** | 8.01 | 19.08 | .04 | **4.16** | 2.16 | 6.16 | 376.4 | .19 |
| Romania | .05 | **7.82** | 4.90 | 10.74 | .52 | **36.16** | 27.63 | 44.69 | .04 | **4.29** | 1.17 | 7.41 | 431.8 | .25 |
| Indonesia | .12 | **16.15** | 12.04 | 20.27 | .16 | 3.51 | −2.13 | 9.16 | .04 | **4.33** | 2.18 | 6.47 | 350.2 | .08 |
| Georgia | .05 | **7.43** | 3.71 | 11.15 | .30 | **15.28** | 6.31 | 24.25 | .04 | **4.44** | 1.15 | 7.73 | 395.1 | .10 |
| El Salvador | .06 | **8.10** | 4.39 | 11.80 | .22 | **8.38** | 3.00 | 13.77 | .04 | **4.50** | 2.46 | 6.53 | 356.9 | .17 |
| Peru | .14 | **18.34** | 14.71 | 21.98 | .28 | **12.89** | 7.03 | 18.75 | .04 | **4.62** | 2.42 | 6.83 | 379.8 | .19 |
| Brazil | .17 | **22.36** | 19.24 | 25.48 | .27 | **11.82** | 6.43 | 17.21 | .05 | **4.78** | 2.81 | 6.75 | 347.8 | .19 |
| Norway | .08 | **10.50** | 6.61 | 14.38 | .37 | **20.77** | 10.10 | 31.44 | .05 | **4.84** | 1.02 | 8.66 | 431.8 | .11 |
| Panama | .09 | **11.97** | 8.31 | 15.63 | .25 | **9.87** | 4.17 | 15.58 | .05 | **4.98** | 2.59 | 7.37 | 358.3 | .21 |
| North Macedonia | .05 | **7.36** | 4.49 | 10.23 | .35 | **18.74** | 12.26 | 25.21 | .05 | **5.09** | 2.57 | 7.60 | 388.9 | .13 |
| Jamaica | .13 | **16.85** | 13.11 | 20.60 | .22 | 6.61 | −2.65 | 15.87 | .05 | **5.12** | 1.38 | 8.85 | 361.3 | .11 |
| Kazakhstan | .06 | **9.50** | 7.45 | 11.55 | .19 | 1.90 | −4.53 | 8.34 | .06 | **5.84** | 3.66 | 8.02 | 406.3 | .05 |
| Latvia | .08 | **11.94** | 9.09 | 14.79 | .37 | **16.49** | 6.67 | 26.31 | .06 | **6.51** | 2.95 | 10.07 | 453.3 | .14 |
| Thailand | .15 | **19.93** | 15.68 | 24.18 | .24 | 1.43 | −4.82 | 7.68 | .07 | **7.68** | 5.13 | 10.24 | 370.6 | .13 |
| Iceland | .06 | **10.07** | 4.63 | 15.52 | .34 | 11.03 | −2.98 | 25.04 | .07 | **7.69** | 2.65 | 12.73 | 429.8 | .10 |
| Brunei Darussalam | .12 | **17.57** | 14.74 | 20.40 | .38 | **10.68** | 3.10 | 18.25 | .09 | **9.00** | 6.11 | 11.88 | 408.8 | .19 |
| United Arab Emirates | .10 | **15.37** | 13.77 | 16.97 | .39 | 3.35 | −2.04 | 8.74 | .12 | **12.40** | 10.22 | 14.59 | 390.2 | .11 |
| Malaysia | .06 | **10.60** | 6.00 | 15.20 | .36 | −1.07 | −6.95 | 4.81 | .12 | **12.96** | 10.06 | 15.85 | 405.9 | .20 |
| Qatar | .02 | **7.05** | 4.25 | 9.85 | .44 | 3.87 | −4.76 | 12.51 | .13 | **13.89** | 10.35 | 17.42 | 403.7 | .16 |

Note: All statistics were pooled across 10 plausible values and weighted using W_FSTUWT.

β represents standardized coefficients, while B refers to unstandardized coefficients.

Statistically significant coefficients (p < .05) were presented in bold.

The countries were sorted in ascending order of the interaction term's coefficient.

[a]The results refer only to 18 out of 27 regions.

between low- and high-socioeconomic status groups. Bars shaded in darker colors indicate that the interaction between growth mindsets and socioeconomic status was statistically significant.

As shown in Fig 2, the seven countries with significant negative interactions showed that students from low socioeconomic backgrounds benefited more from growth mindsets than their peers from high socioeconomic backgrounds, with gaps favoring the low-socioeconomic group ranging from 6.16 points in Italy to 19.96 points in Singapore. For instance, in Singapore, a socioeconomically advantaged student with a growth mindset would score 18.23 points higher in math

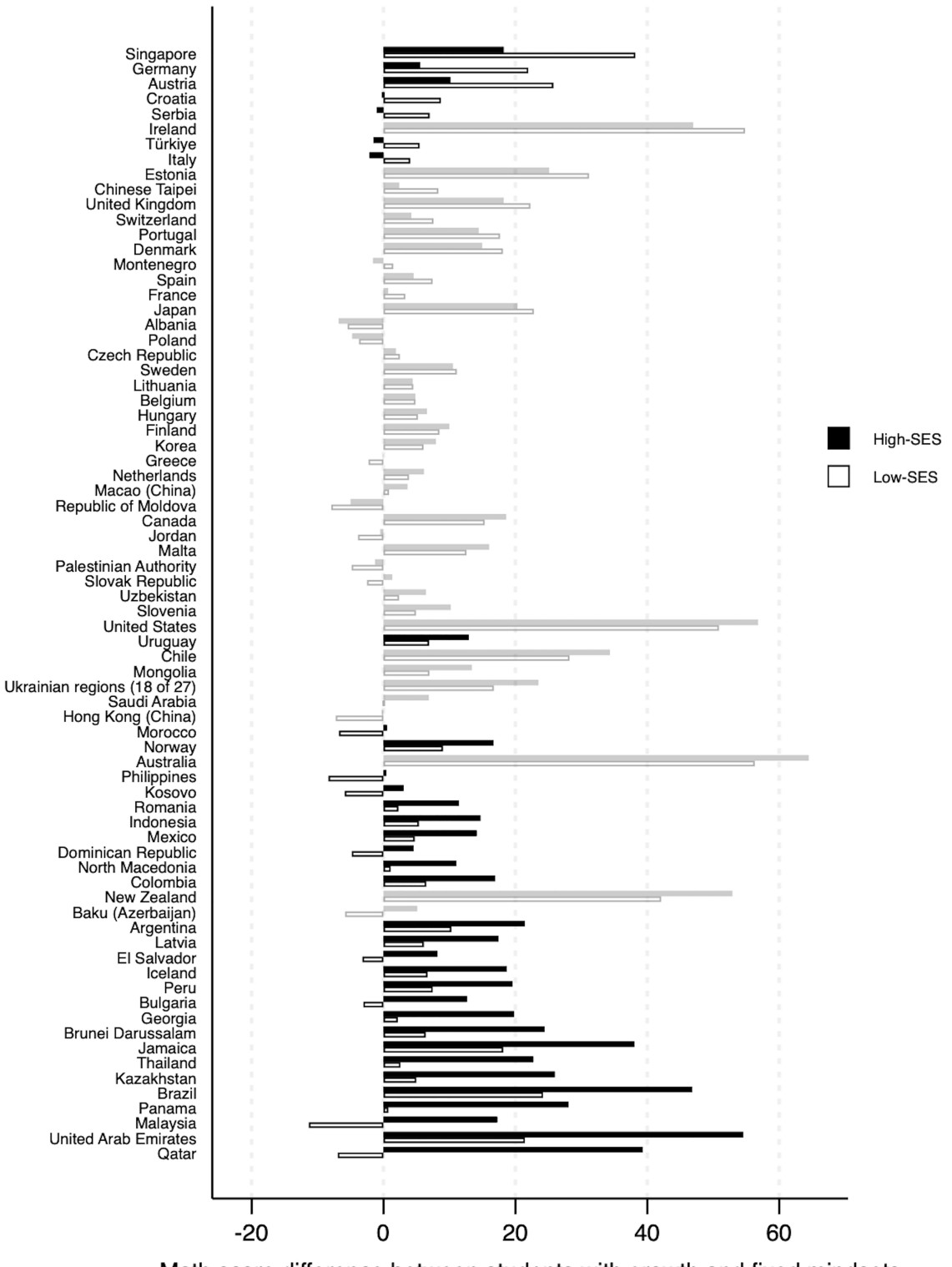

**Fig 2. Score-point difference between students holding growth and fixed mindsets by countries.** Note: The low and high values of socioeconomic status (SES) and growth mindsets were computed at the 16th and 84th percentiles, respectively. Darker-toned bars indicate a statistically significant interaction between growth mindsets and SES (p < .05).

than a comparable advantaged student with a fixed mindset, while a socioeconomically disadvantaged student with a growth mindset would score 38.19 points higher in math than a comparable disadvantaged student with a fixed mindset. A difference of 19.96 points in PISA math achievement scores is meaningful and substantial, especially given that the overall average math score in PISA 2022 declined by 15 points compared to PISA 2018, a drop that is believed to be partly due to the impact of the COVID-19 pandemic [46]. In contrast, 26 countries showed significant positive interactions, where growth mindsets were more beneficial for students from higher socioeconomic backgrounds, with gaps favoring the high-socioeconomic group ranging from 6.01 points in Uruguay to 46.24 points in Qatar.

### Robustness analyses

Following the main analyses, we assessed the robustness of the interaction between growth mindsets and socioeconomic status by estimating three alternative model specifications. Information on covariates and the detailed discussion are reported in S2 Appendix. The first alternative model (Model 3), which controlled for students' gender and immigrant status, yielded minor changes of standardized coefficients of less than ±0.01 compared to the baseline model (Model 2). The results for each country are available in S1 Table. These two demographic backgrounds are typically utilized as covariates when predicting PISA achievement, given their potential influence on academic outcomes [12,67].

In addition, we included three non-cognitive predictors (math self-efficacy, math anxiety, and math proactive behaviors) into Model 4. These non-cognitive factors were selected as previous studies found strong relationships between them and math achievement [68,69]. After adding these variables, the changes in standardized estimates were generally within ±0.01 to ±0.04 of the baseline model in most countries (see S2 Table).

The third specification (Model 5) incorporated a two-way interaction of growth mindsets with gender and immigrant status, while controlling for the three non-cognitive abilities from the previous specification (see S3 Table). The changes in the estimates of Model 5 were also approximately within ±0.01 to ±0.04 of the baseline model in most countries. The summarized changes between each model compared to the baseline can be found in S4 Table. Overall, the robustness test results suggested that the moderating role of socioeconomic status in the association between growth mindsets and math achievement remained robust across alternative specifications.

## Discussion

Table 4 summarizes the key findings from the regression and moderation analyses. While the results varied across countries, growth mindsets were positively associated with math achievement in over two-thirds of the PISA 2022 participating countries. Additionally, socioeconomic status moderated this association in nearly half of the countries, with the majority showing that students from higher socioeconomic backgrounds gained greater benefits from having growth mindsets compared to their peers from lower socioeconomic backgrounds.

**Table 4. Summary of the results.**

| Research question | Results | Number of countries | % of total countries (N = 74) |
|---|---|---|---|
| RQ1 | Positive association between growth mindsets and math achievement. | 50 | 67.6 |
| | Negative association between growth mindsets and math achievement. | 5 | 6.8 |
| | No association between growth mindsets and math achievement. | 19 | 25.7 |
| RQ2 | Higher-SES students gained greater achievement benefits. | 26 | 35.1 |
| | Lower-SES students gained greater achievement benefits. | 7 | 9.5 |
| | SES did not moderate the mindset-achievement association. | 41 | 55.4 |

Note: RQ1 investigated the association between growth mindsets and math achievement after controlling for socioeconomic status. RQ2 examined whether socioeconomic status moderated the association between growth mindsets and math achievement. SES = socioeconomic status.

## Growth mindsets and math achievement

Our findings indicated that the association between growth mindsets and math achievement varied widely across countries. The association consisted of three patterns: positive, negative, and null effects. First, in around two-thirds of the participating countries, students with growth mindsets outperformed their counterparts with fixed mindsets. Growth mindsets have the most positive predictive effects on students' math achievement in Australia, the United States, Ireland, New Zealand, and the United Kingdom, respectively. While the United States is the origin of mindset theory, the other four nations have also been active in research and implementation of mindset interventions over the past decades [70,71]. These countries share several common characteristics, including being predominantly English-speaking, Western, high-income, and developed economies. This finding aligns with the idea that growth mindsets may be more beneficial in affluent countries [36].

Conversely, significant negative associations were found in the Palestinian Authority, the Philippines, Hong Kong, Poland, and the Republic of Moldova. These five countries do not share clear-cut geographical or cultural patterns. Except for the Palestinian Authority, which participated in PISA for the first time in 2022, the predictive effect of growth mindsets on math achievement in four other countries was previously found to be either non-significant or even positive in 2018 [22,30]. While the recession occurred in most participating countries in PISA 2022, the changes observed in these four particular cultural groups may warrant further investigation.

Regional variations in the benefits of growth mindsets were evident in Central Europe and East Asia. In Central Europe, although the predictive role of growth mindsets was positively significant in Hungary, Germany, Austria, and Slovenia, this finding did not hold true in the Czech Republic, the Slovak Republic, and Switzerland. In Poland, growth mindsets were observed to be negatively related to math achievement. Likewise, our findings were in contrast with suggestions from the previous PISA 2018 report [22], which argued that growth mindsets might not be as effective in East Asia as they are in the West. We found that growth mindsets were positively associated with students' math achievement in South Korea, Japan, and Chinese Taipei. However, the association was not found in Macao, while a significant negative association was observed in Hong Kong. These findings suggest that the predictive role of growth mindsets in academic achievement varies across countries, even among those that are geographically close or share similar cultural backgrounds.

## Growth mindsets and achievement across socioeconomic status

Socioeconomic status may moderate the degree to which growth mindsets are associated with academic achievement [8,31,35,37]. Our findings showed three distinct patterns: (1) the association between growth mindsets and academic achievement was universal across socioeconomic backgrounds, (2) the association was stronger among students from higher socioeconomic backgrounds, and (3) the association was stronger among students from lower socioeconomic backgrounds.

First, the association between growth mindsets and math achievement was not different across students with different socioeconomic levels in more than half of the PISA 2022 participating countries, including Chile and the United States. Our findings diverged from a seminal study in Chile [8], which found that students from low-income households experienced greater benefits from growth mindsets in math achievement. Both our study and their study focused on a similar age group. However, the discrepancy in findings may be attributed to differences in the tools used to measure growth mindsets, socioeconomic status, and math achievement. For example, socioeconomic status in the PISA data was calculated from several domains of students' families (e.g., parental educational attainment and domestic possessions), while that of the study by Claro et al. [8] only refers to the family income decile.

In the United States, the existing literature has shown mixed findings. In our study, we found that students benefited from growth mindsets regardless of their socioeconomic backgrounds. This finding aligns with the studies by Destin et al. [38] and Wang et al. [72] which found no evidence of an interaction between growth mindsets and socioeconomic status

in predicting achievement. In contrast, the study by King and Trinidad [37] found that the growth mindsets were negatively related to math achievement for the low socioeconomic status group, whereas a positive association was observed for the high socioeconomic status group. All of these studies used large-scale samples of adolescents approximately 15 years old. These inconsistencies suggest that socioeconomic status may not be a strong moderator of growth mindset effects, or that other contextual factors may play a more significant moderating role. For instance, a recent study involving a nationally representative sample of high school students in the United States found that a growth mindset intervention was more effective for students whose teachers held growth mindsets [73].

Second, in approximately one-third of the countries, students from high-socioeconomic status backgrounds derived greater benefits from having growth mindsets. This trend exhibited a geographical pattern, particularly in Latin America and Southeast Asia, except for Singapore. Our finding aligns with prior studies, which suggested that the effects of growth mindsets on academic achievement were more positive among high-socioeconomic status students in the Philippines [35] and some other Southeast Asian countries [22,34].

Based on mindset research, beliefs about intelligence shape students' meaning systems, leading to more positive views regarding setbacks, learning goals, and efforts to improve [3]. This then influences their actions, such as working harder, taking more time, and seeking help from others. The process is believed to result in academic achievement [3,5]. One crucial component regulating the association between growth mindsets and achievements could be students' abilities or opportunities to perform actions. Students may value the difficulties and desire to act accordingly, but if they do not have access to resources, the process cannot be fully completed. In other words, mindsets do not necessarily translate into actual improvement when people are restricted by limited resources to behave as they prefer [39]. Based on our findings, there was a pattern showing that growth mindsets may be more beneficial for socioeconomically advantaged students in some developing countries with critical wealth equality gaps. This notion was supported by the similar trend observed in Latin America, Southeast Asia, and some European nations with substantial inequality.

Finally, in about 10% of the countries, such as Singapore, Germany, and Austria, growth mindsets tended to have greater benefits for math achievement among low socioeconomic status students. In these countries, students from low socioeconomic backgrounds may have substantially equitable access to educational support systems, which enable those with growth mindsets to respond to academic setbacks independently. For example, in Singapore, the government has made efforts to reduce educational disparities by encouraging initiatives that provide public and community-based tutoring programs for disadvantaged students [74]. Similarly, in Austria and Germany, the educational system concentrated on providing equal autonomy over the curriculum and resource management across public and private schools [51], which could reduce disparities in access to good-quality educational supports. In such conditions, students from low-socioeconomic families who hold growth mindsets may be able to capitalize on their learning goals and efforts, as they could access a range of adequate support, resulting in better measurable academic gains. These findings suggest that the growth mindset may not simply exert desirable benefits on its own for all students, but rather requires supportive conditions and contexts tailored for a particular group of students. Likewise, the growth mindset and socioeconomic status interact differently across countries, even when the countries are geographically located in the same region. Therefore, it is crucial to be cautious when applying broad generalizations.

## Limitations and future directions

Several limitations should be considered. First, the analysis is based on correlational data, which precludes any causal inferences about the relationship between growth mindset, socioeconomic status, and math achievement. Future research should therefore employ longitudinal or experimental designs to support causal links and better understand underlying mechanisms. Second, as the study relies on secondary data from PISA 2022, the selection of variables was limited to those already included in the dataset, restricting the ability to explore other potentially relevant factors. Future studies may consider involving contextual factors that contain ecological influences, such as the mindsets of parents, teachers, and

peers [39]. Likewise, the SES indicator was limited to PISA's definition, which may not capture all aspects of SES in social science research; hence, future research should also involve other alternative indices of SES. Third, the findings are specific to 15-year-old students from countries that participated in PISA 2022 and may not be generalizable to younger or older populations or to educational contexts outside the participating countries. Future investigations should include a broader range of subject domains, age groups, and countries to improve the generalizability and applicability of the findings.

Additionally, we assessed growth mindsets using a single-item measure. The association between this item and various academic outcomes has been observed [30]. However, it may not fully capture the complexity of the construct, and information regarding its internal consistency was not available. Future work may use multi-item measures to improve construct validity and reliability. Moreover, this study concentrated on the relations between mindset and math achievement. The results cannot be generalized to other subjects. Future research should replicate our study using other core subjects of PISA and other types of educational outcomes, such as self-efficacy or anxiety regarding math [75]. At the same time, another limitation to be noted concerns the relationship between math literacy captured in PISA. A recent empirical study argued that the test scores in PISA accounted for general cognitive ability of traditional intelligence tests rather than for domain-specific abilities [55]. Future studies may examine the relationship between growth mindsets and academic achievement measured by large-scale evaluations of student achievement, which are closely tied to specific knowledge based on the national curriculum.

## Conclusions

This study examined the relationship between growth mindsets, socioeconomic status, and math achievement in 74 countries. Our findings suggest that students in most countries benefit from holding the belief that intelligence is malleable. Moreover, socioeconomic status moderated the extent to which math achievement was predicted by growth mindsets in nearly half of these countries. However, the magnitude and direction of this moderation varied. In some countries, students from low-socioeconomic backgrounds benefited more from adopting a growth mindset, whereas in many others, it was the opposite. These findings indicate that the effects of growth mindsets are not universal, but rather shaped by country-specific factors. This variation underscores the importance of promoting growth mindsets through a culturally responsive lens. Socioeconomic conditions unique to each country may shape the extent to which mindset beliefs influence academic outcomes. Therefore, to support inclusive academic achievement, efforts to foster growth mindsets should be carefully tailored to the specific circumstances of students in each context.

## Supporting information

**S1 Appendix. Moderation plots of the association between growth mindsets and math achievement at high and low socioeconomic status levels by country.**
(PDF)

**S2 Appendix. Robustness analyses.**
(DOCX)

**S1 Table. Model 3 Moderation analysis for math literacy controlling for demographic variables.**
(XLSX)

**S2 Table. Model 4 Moderation analysis for math literacy controlling for demographic and non-cognitive variables.**
(XLSX)

**S3 Table. Model 5 Moderation analysis for math literacy with two-way interactions between growth mindsets and demographic variables.**
(XLSX)

**S4 Table. The summary of the results from robustness testing.**
(XLSX)

## Author contributions

**Conceptualization:** Pimmada Charoensilp, Suppanut Sriutaisuk.

**Data curation:** Pimmada Charoensilp.

**Formal analysis:** Pimmada Charoensilp.

**Funding acquisition:** Suppanut Sriutaisuk.

**Investigation:** Pimmada Charoensilp, Suppanut Sriutaisuk.

**Methodology:** Pimmada Charoensilp, Hanjoe Kim, Suppanut Sriutaisuk.

**Resources:** Suppanut Sriutaisuk.

**Supervision:** Suppanut Sriutaisuk.

**Visualization:** Pimmada Charoensilp.

**Writing – original draft:** Pimmada Charoensilp.

**Writing – review & editing:** Pimmada Charoensilp, Hanjoe Kim, Suppanut Sriutaisuk.

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
