## [Decision Letter · Decision Letter 0]

21 Aug 2025

PLOS ONE

Dear Dr. Sriutaisuk,

Thank you for submitting your manuscript to PLOS ONE. After careful consideration, we feel that it has merit but does not fully meet PLOS ONE’s publication criteria as it currently stands. Therefore, we invite you to submit a revised version of the manuscript that addresses the points raised during the review process.

We look forward to receiving your revised manuscript.

Kind regards,

Goutam Saha, PhD

Academic Editor

PLOS ONE

Journal Requirements:

“This research project is supported by Chulalongkorn University, the Second Century Fund (C2F), and has been granted funds from East West Psychological Science Research Center, Faculty of Psychology, Chulalongkorn University.”

3. Thank you for uploading your study's underlying data set. Unfortunately, the repository you have noted in your Data Availability statement does not qualify as an acceptable data repository according to PLOS's standards.

Additional Editor Comments:

I suggest that authors move the results presented in Appendices S1, S2, and S3 into the main manuscript.Also, add a discussion of the results for each of the appendices (S1, S2, and S3).Please provide the data sources for each country in the new Appendix S5.Please follow the manuscript preparation guidelines provided on the PLOS ONE website.

Reviewers' comments:

Reviewer's Responses to Questions

**Comments to the Author**

1. Is the manuscript technically sound, and do the data support the conclusions?

Reviewer #1: Partly

Reviewer #2: Yes

2. Has the statistical analysis been performed appropriately and rigorously?

Reviewer #1: No

Reviewer #2: Yes

3. Have the authors made all data underlying the findings in their manuscript fully available?

Reviewer #1: Yes

Reviewer #2: Yes

4. Is the manuscript presented in an intelligible fashion and written in standard English?

Reviewer #1: Yes

Reviewer #2: Yes

Reviewer #1: This is an interesting and partly well-crafted study. It is mostly clearly written and provides a background of the research problem and topic. Statistical analyses support some of the conclusions and provide contextual nuances with regard to the findings - there are indeed many countries that do not conform to the hypotheses. Nevertheless, there are many things that need to be corrected or improved prior to potential publication. A lot of research has been published based on PISA data so it is important to publish research which is less mediocre and provdes reliable findings.

1. More empirical research should be included. Use Google Scholar and search for the years 2024-2025 so you do not miss recent papers. Here are some suggestions.

https://link.springer.com/article/10.1007/s43545-024-00955-0

https://link.springer.com/article/10.1007/s10212-024-00805-w

https://journals.sagepub.com/doi/full/10.1177/21582440231225870

2. Lack of theoretical background, in part. If countries differ much with regard to these findings, why is that the case? It should come as little surprise when one reads te results. Hence, you have to improve the theoretical backgriund in that regards, particularly with regard to contextual and cultural factors.

3. References to important aspects of SES is one of the main weaknesses of this MS. It is far from enough to just mention the SES index in PISA, which has been criticized in many respects. Here are some important papers to consider:

Avvisati, F. (2020). The measure of socio-economic status in PISA: a review and some suggested improvements. Large-scale Assessments in Education, 8.

Marks, G. N., & O’Connell, M. (2021). Inadequacies in the SES–achievement model: Evidence from PISA and other studies. Review of Education, 9(3).

Wiberg, M., & Rolfsman, E. (2023). Students’ Self-reported Background SES Measures in TIMSS in Relation to Register SES Measures When Analysing Students’ Achievements in Sweden. Scandinavian Journal of Educatioal Research, 67(1), 69–82.

But not only that - you have to explain to readers what SES indicators are typically used in social science research and what the PISA SES index taps into and not.

4. Lack of standardized beta coeffcients. One of the quantitative flaws of the MS is the lack of standardized beta coeffcients, which are very important in educational and psychological research, even if some measures are on the same scale. If it is too burdensome to add this for all countries, you have to do it for at least 10 countries as a robustness check and compare the effect sizes. You may use IDB Analyzer for this as an option to Stata. But Stata works, too.

5. My fifth point is connected to 3 and 4 - you must perform substantial robustness analyses, using appropriate software. One of the key elements of multivariate analyses is control factors to not omit potentially influential factors. Such variables include gender, migration background (which may be separated from the SES index), and other non-cognitive abilities such as self-efficacy, perserverance (perhaps linked to Big Five conscientiousness according to some measures). Only then can have some confidence in your findings. Interaction variables can be included in the same multiple regression models - and make sure to include the standardized effect sizes for comparability. If you use some software you can also incorporate BIC values.

https://psycnet.apa.org/record/2024-86110-001

6. Lastly, you may add a discussion about growth mindset, PISA, cognitive abilities and self-rated abilities (often categorized as generic or domain-specific academic self-concept). This may be incorporated into the theoretical section. It is important to note that PISA is not a usal ILSA but something which substantially taps into cognitive skills and abilities, not that different from the g-factor of conventional intelligence tests. Even if unintentional, that is what empirical analyses whow (see particularly Pokropek et al., 2022).

Pokropek A, Marks GN, Borgonovi F (2022) How much do students’ scores in PISA reflect general intelligence and how much do they reflect specific abilities? J Educ Psychol 114(5):1121–1135

Furnham A, Cheng H (2024) The role of parents, teachers, and pupils in IQ test scores: correlates of the programme for international student assessment (PISA) from 74 countries. Pers Individ Dif 219

Reviewer #2: This study is an exceptionally well-documented and original work that makes a substantial contribution to the international literature on the relationship between growth mindsets, mathematics achievement, and the role of socioeconomic background. Using the rich and consistently reliable PISA 2022 dataset, the authors manage to analyze trends across 74 countries, offering both a global and a detailed perspective.

The research stands out for:

The scope and representativeness of the sample, which provides strong external validity to the conclusions.

Methodological rigor, employing statistical techniques that account for the complexity of the PISA design and the plausible values of achievement.

A critical and balanced presentation of results, highlighting positive, negative, and non-significant findings, while avoiding oversimplifications.

The emphasis on cultural and contextual differences, underscoring the need for tailored educational interventions rather than one-size-fits-all solutions.

The discussion of the findings successfully links empirical evidence with existing theory, while also acknowledging limitations and proposing clear directions for future research. The article not only deepens the understanding of how a growth mindset relates to mathematics achievement internationally, but also offers practical implications for education policies aiming to reduce inequalities.

Overall, this is a carefully crafted, scientifically rigorous, and socially relevant study that can serve as a reference point for researchers, educators, and policymakers alike.

**Do you want your identity to be public for this peer review?** For information about this choice, including consent withdrawal, please see our Privacy Policy

Reviewer #1: No

Reviewer #2: No

---

## [Author Response · Author response to Decision Letter 1]

12 Oct 2025

Manuscript PONE-D-25-33938

Response to Reviewers

Dear Dr. Saha,

We sincerely thank you and reviewers for taking the time to provide such thoughtful and constructive feedback on our manuscript entitled “Relationships between growth mindsets and math achievement across socioeconomic status in 74 countries: Evidence from PISA 2022.” We have carefully considered all comments and made appropriate revisions to our manuscript. Below, we provide a point-by-point response to the editors’ and reviewers’ comments and concerns. References cited in this response are listed at the end.

Editorial and Journal Requirements:

Comment 1: Please ensure that your manuscript meets PLOS ONE's style requirements, including those for file naming. The PLOS ONE style templates can be found at

Author Response: Thank you for providing the URLs to assist us in revising the manuscript. We have ensured that our manuscript adheres to PLOS ONE’s style requirements.

Comment 2: Thank you for stating the following financial disclosure:

“This research project is supported by Chulalongkorn University, the Second Century Fund (C2F), and has been granted funds from East West Psychological Science Research Center, Faculty of Psychology, Chulalongkorn University.”

Author Response: We confirm that the funders had no role in the study. Therefore, we have included the amended Role of Funder statement in the cover letter.

Comment 3: Thank you for uploading your study's underlying data set. Unfortunately, the repository you have noted in your Data Availability statement does not qualify as an acceptable data repository according to PLOS's standards.

Author Response: We have uploaded the minimal dataset necessary to replicate our study’s findings in Stata format (.dta) and shared the Stata analysis script on the Open Science Framework platform at https://osf.io/b2a6k/?view_only=9a352016b926445f8130b5ad18eb9d56.

Comment 4. Please include your full ethics statement in the ‘Methods’ section of your manuscript file. In your statement, please include the full name of the IRB or ethics committee who approved or waived your study, as well as whether or not you obtained informed written or verbal consent. If consent was waived for your study, please include this information in your statement as well.

Author Response: We have added the full name of the IRB in the ‘Methods’ section (Lines 219-221, p. 12).

Comment 5: Please include captions for your Supporting Information files at the end of your manuscript, and update any in-text citations to match accordingly. Please see our Supporting Information guidelines for more information: http://journals.plos.org/plosone/s/supporting-information.

Author Response: We apologize for the earlier oversight. In the revised manuscript, we have added captions for all Supporting Information files at the end of the manuscript and updated the corresponding in-text citations accordingly.

Comment 6: If the reviewer comments include a recommendation to cite specific previously published works, please review and evaluate these publications to determine whether they are relevant and should be cited. There is no requirement to cite these works unless the editor has indicated otherwise.

Author Response: Thank you for the instruction. We have carefully reviewed each of them to evaluate their relevance before deciding whether to cite them.

Additional Editor Comments:

Additional Comment 1: I suggest that authors move the results presented in Appendices S1, S2, and S3 into the main manuscript.

Author Response: Thank you for the suggestion. We have moved the results in Appendices S1, S2, and S3 into the main manuscript as Tables 1, 2, and 3 on pp. 10, 17, and 20, respectively.

Additional Comment 2: Also, add a discussion of the results for each of the appendices (S1, S2, and S3).

Author Response: Discussions of the results of Tables 1, 2, and 3 have been included between lines 302 and 306 (p. 16), lines 315 and 332 (pp. 18-19), and lines 342 and 362 (pp. 19-21), respectively.

Additional Comment 3: Please provide the data sources for each country in the new Appendix S5.

Author Response: We appreciate the editor’s recommendation to provide the data sources for each country. However, because all data for the participating countries are contained in a single dataset available from the official PISA database (https://www.oecd.org/en/data/datasets/pisa-2022-database.html), separate country-level sources are not possible. To ensure clarity, we have added information in the revised manuscript on how to access the publicly available PISA 2022 dataset (Lines 198-200, p. 10) and our cleaned dataset (Lines 281-283, p. 15), as noted in our response to Editorial and Journal Requirements Comment 3 above.

Additional Comment 4: Please follow the manuscript preparation guidelines provided on the PLOS ONE website.

Author Response: We have revised the manuscript to comply with the guidelines provided on the PLOS ONE website.

Reviewers’ Comments to the Authors:

Reviewer #1:

This is an interesting and partly well-crafted study. It is mostly clearly written and provides a background of the research problem and topic. Statistical analyses support some of the conclusions and provide contextual nuances with regard to the findings - there are indeed many countries that do not conform to the hypotheses. Nevertheless, there are many things that need to be corrected or improved prior to potential publication. A lot of research has been published based on PISA data, so it is important to publish research that is less mediocre and provides reliable findings.

Author Response: We sincerely appreciate the reviewer’s time and effort in providing thoughtful and constructive feedback on our manuscript. We value the reviewer’s comments and have worked diligently to address all of them. With this revision, we believe that our manuscript is technically sound, and that the data support our conclusions, with the statistical analyses performed appropriately and rigorously.

Comment 1: More empirical research should be included. Use Google Scholar and search for the years 2024-2025 so you do not miss recent papers. Here are some suggestions.

https://link.springer.com/article/10.1007/s43545-024-00955-0

https://link.springer.com/article/10.1007/s10212-024-00805-w

https://journals.sagepub.com/doi/full/10.1177/21582440231225870 (Add more 2024-2025 citations)

Author Response: Thank you for your valuable suggestion to include more recent empirical papers to ensure our work remains up to date and novel. We have searched for papers published between 2024 and 2025 and have now cited 14 empirical papers from this period, 9 of which are newly added in this revision.

Together with the reviewer’s suggested papers, we included these additional articles to:

1. strengthen the rationale of employing the PISA dataset to examine the relationship between growth mindsets, socioeconomic status, and achievement (Lines 162-172, p. 8),

2. discuss that PISA may capture general cognitive abilities (Lines 183-191, p. 9), and

3. discuss the inclusion of covariates used in the robustness analyses between lines 391-412 (pp. 23-24) and in S2 Appendix (see the response to Reviewer 1’s Comment 5).

Comment 2: Lack of theoretical background, in part. If countries differ much with regard to these findings, why is that the case? It should come as little surprise when one reads the results. Hence, you have to improve the theoretical background in that regard, particularly with regard to contextual and cultural factors.

Author Response: We have expanded the theoretical background by including a deeper discussion of how cultural contexts of each country may influence the differences across the education system (Lines 126-141, pp. 6-7). To strengthen this argument, we have also incorporated several pieces of empirical studies regarding mindset theory that examine the effects of macro-environmental factors, such as national economic conditions, cultural orientations, and social axioms, on students’ learning outcomes (e.g., Jia et al., 2021; King & Wang, 2025). We are grateful for this constructive suggestion, which has helped set the theoretical framework for why this study expected country-level variation in the findings.

Comment 3: References to important aspects of SES is one of the main weaknesses of this MS. It is far from enough to just mention the SES index in PISA, which has been criticized in many respects. Here are some important papers to consider:

Avvisati, F. (2020). The measure of socio-economic status in PISA: a review and some suggested improvements. Large-scale Assessments in Education, 8.

Marks, G. N., & O’Connell, M. (2021). Inadequacies in the SES–achievement model: Evidence from PISA and other studies. Review of Education, 9(3).

Wiberg, M., & Rolfsman, E. (2023). Students’ Self-reported Background SES Measures in TIMSS in Relation to Register SES Measures When Analysing Students’ Achievements in Sweden. Scandinavian Journal of Educational Research, 67(1), 69–82.

But not only that - you have to explain to readers what SES indicators are typically used in social science research and what the PISA SES index taps into and not.

Author Response: We appreciate the reviewer for pointing out this crucial aspect of the SES measurement in PISA and suggesting relevant literature. We have expanded the discussion of the criticisms regarding SES in the Measures section (Lines 248-264, pp. 13-14). We have additionally clarified common indicators of SES in social science research to indicate what PISA’s measure can and cannot capture. We also incorporated the suggested literature to strengthen our explanation of the limitations of the PISA SES index (ESCS). In addition, we have added the limitation of the SES measure of PISA in the limitations and recommendations for future studies (Lines 531-533, p. 29).

Comment 4: Lack of standardized beta coeffcients. One of the quantitative flaws of the MS is the lack of standardized beta coeffcients, which are very important in educational and psychological research, even if some measures are on the same scale. If it is too burdensome to add this for all countries, you have to do it for at least 10 countries as a robustness check and compare the effect sizes. You may use IDB Analyzer for this as an option to Stata. But Stata works, too.

Author Response: We fully agree that standardized beta coefficients are highly important in educational and psychological research. It was not burdensome to add them for all countries. Accordingly, we have now included standardized coefficients for all models and all countries. The computation of the standardized coefficients followed established practices in the moderation literature (Hayes, 2022).

To enhance clarity, and following the Editor’s Additional Comment 1, we have moved the two tables presenting regression coefficients from the appendices to the revised manuscript (Tables 2-3) and report the standardized coefficients with a short summary at Lines 315-332 (pp. 18-19) for Model 1 and Lines 342-362 (pp. 19-21) for Model 2. We sincerely thank the reviewer for this valuable feedback, which has helped us improve both the methodological rigor and the presentation of the findings.

Comment 5: My fifth point is connected to 3 and 4 - you must perform substantial robustness analyses, using appropriate software. One of the key elements of multivariate analyses is control factors to not omit potentially influential factors. Such variables include gender, migration background (which may be separated from the SES index), and other non-cognitive abilities such as self-efficacy, perseverance (perhaps linked to Big Five conscientiousness according to some measures). Only then can have some confidence in your findings. Interaction variables can be included in the same multiple regression models - and make sure to include the standardized effect sizes for comparability. If you use some software you can also incorporate BIC values.

https://psycnet.apa.org/record/2024-86110-001

Author Response: We thank the reviewer for this constructive comment, which was greatly helpful. In response, we have conducted substantial robustness tests by adding three additional models (Models 3-5). Specifically, Model 3 (S3 Table) included gender and immigrant status. Model 4 (S4 Table) controlled for three theoretically relevant factors (math anxiety, math self-efficacy, and math proactive behavior). Model 5 (S5 Table) additionally incorporated two-way interactions between growth mindset and gender and immigrant status. A comparison of all models is presented in S6 Table. Overall, the results demonstrated that the standardized coefficients of the “mindset x SES” remained robust across model specifications. These findings indicate that the focused interaction is generally robust.

In the manuscript, the robustness test section is between Lines 391 and 412, pp. 23-24. To clarify the additional models, we have also discussed the information on covariates and detailed results in the S2 Appendix.

Comment 6: Lastly, you may add a discussion about growth mindset, PISA, cognitive abilities, and self-rated abilities (often categorized as generic or domain-specific academic self-concept). This may be incorporated into the theoretical section. It is important to note that PISA is not a usual ILSA but something which substantially taps into cognitive skills and abilities, not that different from the g-factor of conventional intelligence tests. Even if unintentional, that is what empirical analyses show (see particularly Pokropek et al., 2022).

Pokropek A, Marks GN, Borgonovi F (2022) How much do students’ scores in PISA reflect general intelligence and how much do they reflect specific abilities? J Educ Psychol 114(5):1121–1135

Furnham A, Cheng H (2024). The role of parents, teachers, and pupils in IQ test scores: correlates of the Programme for International Student Assessment (PISA) from 74 countries. Pers Individ Dif 219

Author Response: We thank the reviewer for the constructive suggestions and for highlighting relevant literature underscoring the association between PISA test scores and general cognitive abilities. We have carefully reviewed these studies and incorporated them in Lines 183-191, p. 9. We also included an additional note on this matter in the research limitations and recommendations section (Lines 545-550, p. 30) that math achievement, as measured by the PISA framework, may be more closely related to general cognitive abilities rather than domain-specific abilities.

Reviewer #2:

This study is an exceptionally well-documented and original work that makes a substantial contribution to the international literature on the relationship between growth mindsets, mathematics achievement, and the role of socioeconomic background. Using the rich and consistently reliable PISA 2022 dataset, the authors manage to analyze trends across 74 countries, offering both a global and a detailed perspective.

The research stands out for:

The scope and representativeness of the sample, which provides strong external validity to the conclusions.

M

---

## [Decision Letter · Decision Letter 1]

4 Nov 2025

Relationships between growth mindsets and math achievement across socioeconomic status in 74 countries: Evidence from PISA 2022

PONE-D-25-33938R1

Dear Dr. Sriutaisuk,

We’re pleased to inform you that your manuscript has been judged scientifically suitable for publication and will be formally accepted for publication once it meets all outstanding technical requirements.

Kind regards,

Goutam Saha, PhD

Academic Editor

PLOS ONE

Additional Editor Comments (optional):

Reviewers' comments:

Reviewer's Responses to Questions

**Comments to the Author**

Reviewer #1: All comments have been addressed

Reviewer #2: All comments have been addressed

2. Is the manuscript technically sound, and do the data support the conclusions?

Reviewer #1: Yes

Reviewer #2: Yes

3. Has the statistical analysis been performed appropriately and rigorously?

Reviewer #1: Yes

Reviewer #2: Yes

4. Have the authors made all data underlying the findings in their manuscript fully available?

Reviewer #1: Yes

Reviewer #2: Yes

5. Is the manuscript presented in an intelligible fashion and written in standard English?

Reviewer #1: Yes

Reviewer #2: Yes

Reviewer #1: Thank you for a terrific work on this revised version of the manuscript. This version includes more references to relevant literature, more statistical information, sections about limitations about various aspects (e.g., SES indices), as well as robustness checks. All in all, this contributes to a substantially improved article and study.

Reviewer #2: This study is an exceptionally well-documented and original work that makes a substantial contribution to the international literature on the relationship between growth mindsets, mathematics achievement, and the role of socioeconomic background. Using the rich and consistently reliable PISA 2022 dataset, the authors manage to analyze trends across 74 countries, offering both a global and a detailed perspective.

The research stands out for:

•The scope and representativeness of the sample, which provides strong external validity to the conclusions.

•Methodological rigor, employing statistical techniques that account for the complexity of the PISA design and the plausible values of achievement.

•A critical and balanced presentation of results, highlighting positive, negative, and non-significant findings, while avoiding oversimplifications.

•The emphasis on cultural and contextual differences, underscoring the need for tailored educational interventions rather than one-size-fits-all solutions.

The discussion of the findings successfully links empirical evidence with existing theory, while also acknowledging limitations and proposing clear directions for future research. The article not only deepens the understanding of how a growth mindset relates to mathematics achievement internationally, but also offers practical implications for education policies aiming to reduce inequalities.

Overall, this is a carefully crafted, scientifically rigorous, and socially relevant study that can serve as a reference point for researchers, educators, and policymakers alike.

**Do you want your identity to be public for this peer review?** For information about this choice, including consent withdrawal, please see our Privacy Policy

Reviewer #1: **Yes: ** Björn Boman

Reviewer #2: No

---

## [Editor Report · Acceptance letter]

PONE-D-25-33938R1

PLOS ONE

Dear Dr. Sriutaisuk,

I'm pleased to inform you that your manuscript has been deemed suitable for publication in PLOS ONE. Congratulations! Your manuscript is now being handed over to our production team.

Kind regards,

on behalf of

Dr. Goutam Saha

Academic Editor

PLOS ONE